# Revisiting Non-separable Binary Classification and its Applications in Anomaly Detection

## Abstract

The inability to linearly classify XOR has motivated much of deep learning. We revisit this age-old problem and show that *linear* classification of XOR is indeed possible. Instead of separating data between halfspaces, we propose a slightly different paradigm, `equality separation`, that adapts the SVM objective to distinguish data within or outside the margin. Our classifier can then be integrated into neural network pipelines with a smooth approximation. From its properties, we intuit that equality separation is suitable for anomaly detection. To formalize this notion, we introduce *closing numbers*, a quantitative measure on the capacity for classifiers to form closed decision regions for anomaly detection. Springboarding from this theoretical connection between binary classification and anomaly detection, we test our hypothesis on supervised anomaly detection experiments, showing that equality separation can detect both seen and unseen anomalies.

## 1 Introduction

**Linear classification of a non-linearly separable dataset** A common belief in machine learning is that linear models (like the perceptron (Minsky & Papert, 1969)) cannot model some simple functions, as shown by the example XOR (exclusive OR) in classic textbooks (Goodfellow et al., 2016; Hastie et al., 2004). This failure justifies the need for using non-linear transforms such as using neural networks (NNs). We first reexamine this counter-example:

XOR can be learnt by linear models: a single neuron without hidden layers or feature representations.

Taking as input $\mathbf{x} = (x_1, x_2)$ with binary coordinates $x_i \in \{0, 1\}$, we want to learn the logical functions AND, OR, and XOR shown in Figure 1. For the first two configurations AND and OR, it is easy to find a separating line such that half of the space contains only points with label 0 and the other half only points with label 1. However, XOR is not linearly separable, as there is no way to draw a line separating the two classes 0 and 1. Solutions include non-linearities. The most popular is to transform the input via kernels (Seki et al., 2008) or non-linear activation functions with hidden layers (Singh & Pandey, 2016). Figure 1d shows that non-linear feature transforms from $\mathbf{x} = (x_1, x_2)$ to $(h_1, h_2)$ allow for halfspace separation. Conversely, Flake (1993) shows that the *output* representation can be modified non-linearly to solve XOR, but still does not achieve linearity and fails to handle OR.

However, resorting to non-linearities is not necessary. We propose using the line not to separate the input space into 2 halves, but to use it just as a line – a class is either on the line or not. With this, we have a *linear* classifier that can perfectly model the three logical functions above, including XOR (Figure 1). We name this paradigm as *equality separation*. Although equality separation is more conservative than halfspace separation to classify one of the classes, it can solve linearly separable and non-separable problems, demonstrating its flexibility (e.g. Figure 1e, 1f). In fact, we show later that equality separation is more expressive than halfspace separation.

**Anomaly detection** is paramount for detecting system failure. There may be some limited examples of failures (anomalies), but we usually do not have access to all possible types of anomalies. For instance, cyber defenders want to detect known and unknown ('zero-day') attacks. With labeled normal anomaly data, can anomaly detection (AD) be set up as a classic binary classification objective, optimized with standard empirical risk minimization (ERM)? Since equality separation has a smaller positively labeled region than halfspace separation, we posit that equality separation can be used for supervised AD. Particularly, we view points on/close to the line as normal while points far from the line

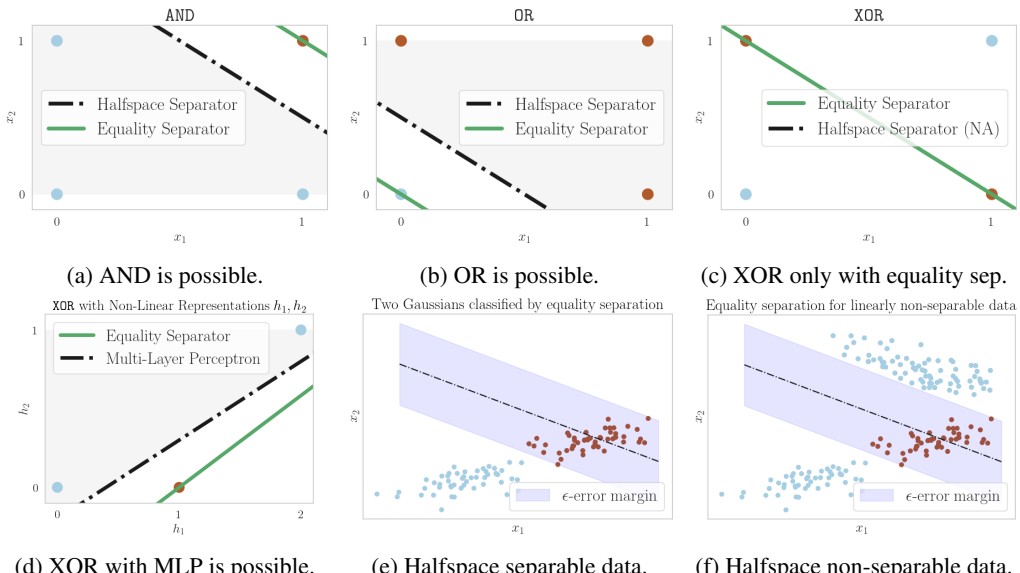

Figure 1: Linear classification by halfspace separators and equality separators for logical functions (Figures 1a-1c) and with 1 hidden layer (Figure 1d). In general, equality separation can classify linearly separable (Figure 1e) and non-separable (Figure 1f) data.

are anomalies, similar to how a scientist screens data sampled from a linear model. To formalize this idea of "suitability", we introduce *closing numbers* to understand a binary classifier's ability to detect anomalies. Based on the suitability of a binary classifier, we can integrate them into NNs for AD.

**Contributions** We propose the *equality separation* paradigm. Equality separators use the distance from a datum to a learnt hyperplane as the classification metric. We summarize our contributions:

- As a linear classifier, they have twice the VC dimension of halfspace separators and can *linearly* classify XOR, which has never been done before. We also propose a smooth approximation for equality separation so that it can be used in NNs and trained with gradient descent.
- We introduce *closing numbers*, a quantitative metric of the ease of classifiers and activation functions to form closed decision regions and their capacity for learning. We corroborate with this theory and empirically observe that equality separation can achieve a balance in detecting both known and unknown anomalies.

## 2 EQUALITY SEPARATOR: BINARY CLASSIFICATION REVISITED

A standard classification problem is as follows. Let $\mathcal{X} \subseteq \mathbb{R}^n$ be the feature space, $\mathcal{Y} := \{0, 1\}$ be the label space and $\mathcal{H}$ be a hypothesis class. Given the true function $c : \mathcal{X} \to \mathcal{Y}$ (i.e. the *concept*), we aim to find hypothesis $h \in \mathcal{H}$ that reduces the generalization error $L(h) := \mathbb{P}(h(\mathbf{x}) \neq c(\mathbf{x}))$ for $\mathbf{x} \in \mathcal{X}$.

Linear predictors (Shalev-Shwartz & Ben-David, 2014, Chapter 9) are the simplest hypothesis classes we can consider. In $\mathbb{R}^n$, the class of linear predictors comprises a set of affine functions

$$A_n = \{\mathbf{x} \mapsto \mathbf{w}^T\mathbf{x} + b : \mathbf{w} \in \mathbb{R}^n, b \in \mathbb{R}\}$$

and a labeling function $\phi : \mathbb{R} \to \mathcal{Y}$. The linear predictor class is

$$\phi \circ A_n = \{\mathbf{x} \mapsto \phi(h_{\mathbf{w},b}(\mathbf{x})) : h_{\mathbf{w},b} \in A_n\}.$$

Halfspace separators as in the perceptron use the sign function $\phi$ to split the space $\mathbb{R}^n$ into 2 with a hyperplane: $\text{sign}(z) = 1$ if $z \geq 0$, and $\text{sign}(z) = 0$ otherwise. Equivalently, $\text{sign}(z) = \mathbb{I}(z \geq 0) = \mathbb{I}_{\mathbb{R}^+ \cup \{0\}}(z)$ for indicator function $\mathbb{I}$. A critical issue in classification is that halfspace separators cannot achieve zero generalization error when the input space $\mathcal{X}$ is linearly inseparable (Minsky & Papert, 1969). To resolve this, we propose to edit the set for the indicator function of $\phi$.

The core change is as follows: we change the labeling from $\phi(z) = \mathrm{sign}(z) = \mathbb{I}(z \geq 0)$ to $\phi(z) = \mathbb{I}(z = 0)$ or $\phi(z) = \mathbb{I}(z \text{ close to } 0)$. We introduce two notions of equality separators: the *strict equality separator* and the generalized, robust $\epsilon$-*error separator*. We define the first type as follows:

**Definition 2.1.** A *strict equality separator* is a decision rule where the hypothesis class is of the form

$$\mathcal{H} = \{\mathbf{x} \longmapsto \mathbb{I}(h_{\mathbf{w},b}(\mathbf{x}) = 0) : h_{\mathbf{w},b} \in A_n\} \cup \{\mathbf{x} \longmapsto \mathbb{I}(h_{\mathbf{w},b}(\mathbf{x}) \neq 0) : h_{\mathbf{w},b} \in A_n\}. \quad (1)$$

Geometrically, the strict equality separator checks if a datum is on or off some hyperplane i.e. $\phi := \mathbb{I}_{\{0\}}$. In halfspace separation, labels are arbitrary, and we capture this with the union of 2 classes in (1).

However, a hyperplane has zero volume and sampled points may be noisy. To increase its robustness, we can allow some error for a datum within $\epsilon$ distance from a hyperplane. To incorporate this $\epsilon$-error margin, we swap mapping $\phi$ from $\mathbb{I}_{\{0\}}$ to $\mathbb{I}_{S_\epsilon}$ for the set $S_\epsilon = \{z \in \mathbb{R} : -\epsilon \leq z \leq \epsilon\}$ for $\epsilon > 0$:

**Definition 2.2.** The $\epsilon$-*error separator* is a decision rule where the hypothesis class is of the form

$$\mathcal{H}_\epsilon = \{\mathbf{x} \longmapsto \mathbb{I}_{S_\epsilon}(h_{\mathbf{w},b}(\mathbf{x})) : h_{\mathbf{w},b} \in A_n\} \cup \{\mathbf{x} \longmapsto \mathbb{I}_{\mathbb{R} \setminus S_\epsilon}(h_{\mathbf{w},b}(\mathbf{x})) : h_{\mathbf{w},b} \in A_n\} \quad \text{for } \epsilon > 0.$$

The $\epsilon$-error separator modifies the SVM objective. In classification, SVMs produce the optimal separating hyperplane such that no training examples are in the $\epsilon$-sized margin (up to some slack) (Cortes & Vapnik, 1995). Meanwhile, the margin in $\epsilon$-error separators is reserved for one class, while the region outside is for the other. With this striking example, we proceed to analyze its expressivity.

## 2.1 VC DIMENSION

Vapnik–Chervonenkis (VC) dimension (Vapnik & Chervonenkis, 1971) measures the complexity of a hypothesis class to give bounds on its generalization error. The VC dimension of a halfspace separator in $\mathbb{R}^n$ is $n + 1$ (Shalev-Shwartz & Ben-David, 2014, Theorem 9.2). We claim that equality separation is not merely an XOR solver, but more expressive than halfspace separators in general:

**Theorem 2.3.** *For hypothesis class of strict equality separators $\mathcal{H}$ in Def. 2.1, $VCdim(\mathcal{H}) = 2n + 1$.*

We defer the proof to Appendix B and build intuition in $\mathbb{R}^2$. We draw a line through the class with the fewest points – the union of indicator function classes allows us to. This is always possible with 5 non-colinear points: the maximum number of points that the class with the minimum points has is 2, which perfectly determines a line. 6 points is not possible: a labeling with 3 points for each class admits an overdetermined system which is unsolvable in general. Extrapolating to higher dimensions, the VC dimension is 1 more than $2n$.

**Corollary 2.4.** *For $\epsilon$-error separators $\mathcal{H}_\epsilon$ in Def. 2.2, $2n + 1 \leq VCdim(\mathcal{H}_\epsilon) \leq 2n + 3$.*

The lower-bound relies on Theorem 2.3 while the upper bound is constructed by the union of hypothesis classes. Details are in Appendix B. We explore the significance of more expressivity in Section 3.4.

**Remark 2.5.** Label flipping is a perspective from which we can understand the doubled VC dimension. Switching labels does not fundamentally change the optimization objective in halfspace separation due to its symmetry. The asymmetry of equality separators yields a different optimization objective if labels are swapped – the goal is to obtain the margin of another class instead. This increases expressivity, making label flipping non-vacuous. For instance, without label flipping, it is impossible to classify the function OR with the strict equality separator (Flake, 1993). However, knowing that OR induces more points from the positive class, we can choose to instead model the negative class.

## 2.2 COMPARISON TO HALFSPACE SEPARATORS AND SVMS

Another intuition of the doubled VC dimension is that the equality separator is the intersection between 2 halfspaces $\mathbf{w}^T \mathbf{x} + b \leq \epsilon$ and $\mathbf{w}^T \mathbf{x} + b \geq -\epsilon$. In contrast, modeling a halfspace separator $\mathbf{w}^T \mathbf{x} + b$ with an equality separator requires non-linearity like ReLU: $\mathbb{I}(\max(0, \mathbf{w}^T \mathbf{x} + b) \neq 0)$.

From a hypothesis testing perspective, equality separators yield a more robust separating hyperplane compared to SVMs in binary classification. We link the $\epsilon$-error separator to the most powerful test, the likelihood ratio test, via Twin SVMs (Jayadeva et al., 2007). In binary classification, Twin SVM learns an $\epsilon$-error separator for each class, $\mathbf{w}_1^T \mathbf{x} + b_1 = 0$ and $\mathbf{w}_2^T \mathbf{x} + b_2 = 0$. For a test datum $\mathbf{x}'$, we can test its membership to a class based on its distance to class $i \in \{1, 2\}$, $\epsilon_i := \frac{|\mathbf{w}_i^T \mathbf{x}' + b_i|}{||\mathbf{w}_i||_2}$. It is

natural to use the ratio between the distances to resemble a likelihood ratio test

$$\mathbb{I}\left(\frac{\epsilon_2}{\epsilon_1} \geq t\right) = \mathbb{I}\left(\frac{|\mathbf{w}_2^T\mathbf{x}' + b_2|}{||\mathbf{w}_2||} - t\frac{|\mathbf{w}_1^T\mathbf{x}' + b_1|}{||\mathbf{w}_1||} \geq 0\right).$$

By constraining the Twin SVM to (1) parallel hyperplanes ($\mathbf{w}_1 = \mathbf{w}_2$), (2) different biases ($b_1 \neq b_2$) and (3) equal importance to classify both classes ($t = 1$), we obtain a separating hyperplane $2\mathbf{w}_1^T\mathbf{x} + b_1 + b_2 = 0$ between the 2 classes (Appendix C). This separating hyperplane lies halfway between the hyperplanes for each class, where $\frac{|b_1 - b_2|}{||\mathbf{w}_1||_2}$ resembles the margin in SVMs. Without loss of generality, for $b_1 > b_2$, the region in between the 2 hyperplanes $\{x \in \mathcal{X} : -b_2 \leq \mathbf{w}_1^T\mathbf{x} \leq -b_1\}$ acts as the soft margin. The separating hyperplane derives from the learnt $\epsilon$-error separators for each class, which is learnt using the mass of each class. This is unlike SVMs, which use support vectors to learn the separating hyperplane (Cortes & Vapnik, 1995). Support vectors may be outliers far from high density regions for each class, so using them may not be as robust as two $\epsilon$-error separators.

## 3 ANOMALY DETECTION

We focus on supervised AD, where anomaly classes are not fully represented in the training data. Distinguishing between the normal and anomaly class is difficult due to (a) class imbalance, (b) lack of common features among anomalies and (c) no guarantees to detect unseen anomalies (Ruff et al., 2020). We aim to find the manifold that normal data (positive class) live on where anomalies do not. This more general non-linear formulation of equality separation shows the utility of our paradigm.

### 3.1 BACKGROUND

**Unsupervised AD** has been the main focus of AD literature, where training data is unlabelled and assumed to be mostly normal (i.e. non-anomalous). Since the anomaly class is not well represented in training, most methods train models on auxiliary tasks using normal data. This assumes that the model is unlikely to perform well at the known task during inference if the datum is unseen/anomalous (Chandola et al., 2009). Popular tasks include regression via autoencoding and forecasting (Kim et al., 2020; Hoffmann, 2007), density estimation (Zong et al., 2018; Breunig et al., 2000) and contrastive learning (Qiu et al., 2021; Shenkar & Wolf, 2022).

**Classification methods** have generally not focused on supervised AD either. Classification tasks that optimize directly on the AD objective artifically generate anomalies (Steinwart et al., 2005; Sipple, 2020; Goyal et al., 2020)), while methods directly solving AD with empirical risk minimization (ERM) do not yield competitive results (Ruff et al., 2020). The lack of common features among anomalies hinders models from generalizing to detecting unseen anomalies. Additionally, synthetic anomalies may be unrepresentative of real anomalies, so covariate shift of the anomaly class during test time may affect AD performance. Nevertheless, directly optimizing over the AD objective is closer to the task and has benefits such as more effective explainable artificial intelligence results (see Sipple (2020)).

**Limited labeled anomalies** is a less explored problem, mostly explored from a semi-supervised approach. Methodologically, this is the same as supervised AD: a small portion of the dataset is labeled while the unlabeled portion is assumed to be (mostly) normal. Pang et al. (2019) proposes deviation networks to aid direct binary classification. They stay within halfspace separation and focus on anomalies limited in *number* rather than *type*. Ruff et al. (2020) proposes Deep Semi-Supervised Anomaly Detection (SAD), which performs an auxiliary task of learning data representations such that normal data have representations within a hypersphere while anomalies fall outside.

**Out-of-distribution (OOD) detection** is the field that motivates our application of equality separation in AD. To reject OOD inputs and reduce approximation error, classifiers can be designed to induce less open decision regions in input space (G.C. Vasconcelos, 1995), also known as open space risk minimization (Scheirer et al., 2013). Decision regions where the decision boundary encloses points belonging to a single class are considered closed. Forming closed decision regions has a strong connection to density level set estimation. For some class with density function $p$, the density level set for significance level $\alpha$ (and corresponding threshold $\tau_\alpha$) can be formulated as (Ruff et al., 2021)

$$C_\alpha := \{\mathbf{x} \in \mathcal{X} \mid p(\mathbf{x}) > \tau_\alpha\}$$

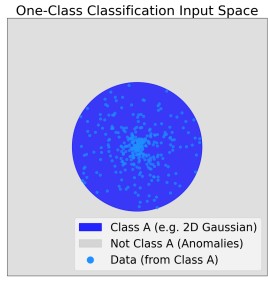 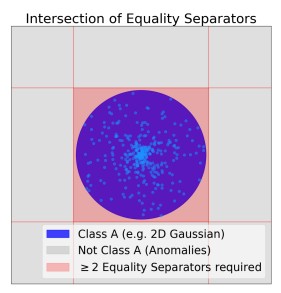 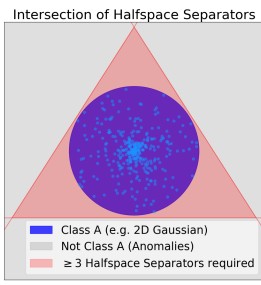

(a) Normal data is in blue circle.   (b) Equality separators: rectangle.   (c) Halfspace separators: triangle.

Figure 2: Normal data occupies a space with non-zero, finite volume. Classifiers can form closed decision regions to capture this.

Table 1: Difference between perceptron, equality separator and RBF network (inputs from $\mathbb{R}^n$).

| Decision Rule | Perceptron | Equality separator | RBF network |
|---|---|---|---|
| Bias in Input Space | Halfspace | Hyperplane | Point/Mean |
| Activation Function | Sigmoid, ReLU | Bump | RBF |
| Properties (Closing Number) | Global ($n+1$) | Semi-local ($n$) | Local (1) |

while anomalies of the class belong to $\mathcal{X} \backslash C_\alpha$. Especially when feature space $\mathcal{X}$ is unbounded, we see a connection between classifiers that induce closed and decision regions and classifiers that induce smaller ('more conservative'), finite-volumed density level sets $C_\alpha$. To model the normal class in (one-class) AD, we can estimate the density level sets. From a simple classification framework, we estimate the level sets from data by forming a decision boundary that encloses points within the class, similar to SAD. Figure 2a is an example where the true decision boundary encloses a region in the input space such that all points in those regions correspond to high density regions of a particular class.

### 3.2 LOCALITY: PRECURSOR TO CLOSED DECISION REGIONS

Using NNs as classifiers can help to find these complex decision boundaries, but it is not immediately clear if the decision regions can be closed and, consequently, if the density level set can be well estimated. In NNs, classifiers are modeled by their smooth versions (activation functions) and have largely been split into 3 categories: global, semi-local and local (Flake, 1993; Hartman & Keeler, 1991). Local mappings, e.g. radial basis function (RBF) networks and activation functions, "activate" around a center point i.e. have outputs beyond a threshold. In contrast, global mappings (e.g. perceptron, sigmoid, ReLU) activate on a whole halfspace. Semi-local mappings straddle in between. In particular, Flake (1993) proposes bump activations, hyper-ridge and hyper-hill classification which are locally activated along one dimension and globally activated along others. Our equality separator casts hyper-ridge and hyper-hill classifiers into linear classifiers with the intuition of activation around a hyperplane. Hence, we view equality separators as semi-local too. We summarize these observations in Table 1.

To model the equality separator as an activation function for gradient-based learning, we seek a smooth function that has a "bump" centered at 0 with a symmetric decay away from 0. In this paper, we use the Gaussian function as our bump activation:

$$B(z, \mu, \sigma) = \exp\left[-\frac{1}{2}\left(\frac{z-\mu}{\sigma}\right)^2\right] \tag{2}$$

where $z$ is the input into the activation function and $\mu, \sigma$ are parameters which can be learnt.

The semi-locality of equality separators and bump activations achieve the benefits of global mappings like sigmoids in perceptrons and local mappings like RBF activations in RBF networks (Flake, 1993). Local activations can converge faster (Hartman & Keeler, 1991) and provide reliability in adversarial settings (Goodfellow et al., 2015; G.C. Vasconcelos, 1995), while global mappings work well in high dimensions with exponentially fewer neurons (Hartman & Keeler, 1991) and are less sensitive to initializations (Flake, 1993), making them easier to train (Goodfellow et al., 2015). Consider our XOR example with a single neuron: sigmoids in perceptrons are too global, while one RBF unit cannot cover 2 points from either class. Equality separation and bump activations limit the globality of the

perceptron but retain a higher globality than RBF activations, which allows them to linearly classify XOR. But what does "globality" or "locality" precisely mean, and can we quantitatively measure this?

### 3.3 LOCALITY AND CLOSING NUMBERS FOR CLOSED DECISION REGIONS

The locality of a classifier not only has implications on classification, but also on AD. Specifically, it is intuitive to see a connection between the locality of classifiers and their ability to form closed decision regions, which in turn is useful for density level set estimation in AD. To quantify the learnability of closed decision regions, we propose one metric to formalize this notion of locality:

**Definition 3.1.** The *closing number* of a hypothesis class is the minimum number of classifiers from the hypothesis class such that their intersection produces closed (positive-volumed[1]) decision region.

The output of a neuron is a linear combination (e.g. scaled weighted average[2]) of neuron activations in the previous layer. Thus, the closing number represents the minimum number of neurons in a layer to induce a closed decision region in the previous layer. The closing number of RBF networks is 1 due to its locality. Meanwhile, the closing number of equality separators is $n$ in $\mathbb{R}^n$, which realizes when the weight vectors of all equality separators are linearly independent. Since one dimension is local for each equality separator, the intersection of $n$ equality separators in $\mathbb{R}^n$ can induce the closed decision region. The closing number of halfspace separators is $n + 1$, which realizes by forming an $n$-simplex volume. In 2D, closed decision regions would correspond to a circle (e.g. the circle in Figure 2a), parallelogram (Figure 2b) and triangle (Figure 2c) respectively. Proofs are in Appendix D.

The implication of closing numbers and the constraints needed for its realization in the continuous version (i.e. activation functions) across adjacent layers in feed-forward NNs is as follows. Closed decision regions are directly induced by RBFs and more easily induced by equality separation (e.g. bump activations) than halfspace separation (e.g. sigmoid or ReLU). We corroborate with Flake (1993) of the semi-locality of bump functions, where it is easier to learn than RBFs but more robust to adversarial settings than halfspace separators (G.C. Vasconcelos, 1995). Additionally, Goodfellow et al. (2015) argued for the decrease robustness of sigmoid and ReLU to adversarial attacks due to the linearity in individual activations. Closing numbers complement this view, providing a global perspective of activations working together to induce closed decision regions. This gives insight on the capacity and learnability of closed decision regions.

### 3.4 CONTROLLING GENERALIZATION ERROR WITH INDUCTIVE BIAS IN AD

Based on the closing number of equality separation and the constraints needed to achieve it, we impose the constraint that the normal class should lie within the margin of equality separators. Using domain knowledge to assign normal data to the positive class, we control the approximation error. Prohibiting label flipping reduces to hyper-hill classification, which has VC dimension of $n + 1$ like halfspace separation (Flake, 1993), controlling the estimation error. Therefore, we focus on AD applications. [3]

## 4 APPLICATIONS

**Optimization scheme**    To obtain a classifier, we perform ERM on data $\{(\mathbf{x}_i, y_i)\}_{i=1}^m$. In this paper, we consider mean-squared error (MSE) and logistic loss (LL) as loss functions. Optimizing with BFGS with LL and MSE on the affine class $A_2$ produces a solution for XOR, corroborating with the results in Flake (1993). For the bump function in (2), we fix $\mu = 0$. A smaller $\sigma$ allows for a stricter decision boundary but limits gradient flow. We defer analysis of the role of $\sigma$ to Appendix F.1.

**Experiments**    For binary classification, we observe that equality separators can classify some linearly non-separable data (e.g. Figure 1f). They are also competitive with SVMs in classifying linearly separable data (e.g. Figure 1e, where equality separation is more conservative in classifying the positive brown class). Since our focus is to evaluate in supervised AD, we defer results and further

---

[1]Meaningful analysis requires constraints that decision regions (in raw or latent space) has non-zero volume.

[2]Parameters can be normalized while negations represent the complement of a region.

[3]We note that AD is hard because any validation set has no information about unseen anomalies, so its utility as a proxy of test-time generalization is limited. Thus, generalization heavily relies on the design of a model.

Table 2: Test AUPR for synthetic data comparing shallow models and neural networks.

(a) Shallow models.

| Shallow Models | AUPR |
|---|---|
| DT | 0.87±0.03 |
| RF | 0.88±0.01 |
| XGB | 0.55±0.01 |
| LR | 0.54±0.00 |
| SVM | 0.54±0.00 |
| OCSVM | 0.83±0.00 |
| IsoF | 0.86±0.05 |
| LOF | **1.00±0.00** |
| ES (ours) | **1.00±0.00** |

(b) For NNs, we modify the output layer (halfspace separation (HS), equality separation (ES) and RBF separation (RS)) in the rows and activation functions in the columns. Random classifier AUPR is 0.75.

| NN\Activation | Leaky ReLU | Bump | RBF |
|---|---|---|---|
| HS | 0.62±0.12 | 0.92±0.13 | 0.83±0.18 |
| ES (ours) | 0.83±0.15 | 0.99±0.04 | **1.00±0.00** |
| RS | 0.97±0.05 | 0.91±0.14 | 0.97±0.06 |

discussion of each experiment to Appendix F.2. For supervised AD, we explore 2 settings. The first is with synthetic data, where we can understand the differences between classifiers more clearly and perform ablation studies. The second is with real-world cyber-attacks. We evaluate the ability of models to separate normal and anomaly classes with area under the precision-recall curve (AUPR).

## 4.1 SUPERVISED ANOMALY DETECTION: TOY MODEL

We explore NNs as embedding functions into a linear feature space for non-linear supervised AD. With a simple binary classification objective, we test how well different NNs can form one-class decision boundaries. Other deep methods have been used for unsupervised (Zong et al., 2018), semi-supervised (Ruff et al., 2020) and supervised (Yamanaka et al., 2019) AD. We only consider directly optimizing over the traditional binary classification objective.

**Setup**  To demonstrate AD with seen and unseen anomalies, we generate synthetic 2D data by adapting the problem in Sieradzki et al. (2022). The normal (positive[4]) class is uniformly sampled from a circle of radius 1, and the anomaly (negative) class during training and testing is uniformly sampled from circles of radius 2 and 0.5 respectively, with all centred at the origin. This task is not trivial because it is easy to overfit the normal class to the ball rather than just its boundary. To challenge each method, we limit this experiment to 100 data samples. More details can be found in Appendix F.3.

**Models**  We make apples-to-apples comparisons with classical binary classifiers: decision trees (DT), random forest (RF), extreme gradient boosted trees (XGB) and halfspace methods logistic regression (LR) with RBF kernel and SVM with RBF kernel. To illustrate the difficulty of this AD task, we also use standard AD baselines: One-Class SVM (OCSVM) with RBF kernel, Local Outlier Factor (LOF) and Isolation Forest (IsoF).[5] We compare these shallow models with equality separators with an RBF kernel. We also compare NNs directly trained for binary classification. We modify NNs with 2 hidden layers to have (a) halfspace separation, RBF separation or equality separation at the output layer, and (b) leaky ReLU, RBF or bump activations. We provide quantitative (Table 2) and qualitative (Figure 3) analyses that suggest either accurate one-class classification or overfitting to seen anomalies.

**Shallow models**  From Table 2, we see classical binary classifiers XGB, LR and SVM perform worse than random and are inappropriate for supervised AD. DT, RF and LOF perform better than random but overfit the training data; even though LOF has a perfect test AUPR, it has a train AUPR of 0.84, suggesting a bad one-class decision boundary (Figure 3c), while DT and RF overfit with an $\ell_\infty$ norm-like decision boundary (e.g. Figure 3a). IsoF has a high AUPR (Figure 3b), but our ES with RBF kernel has the highest and perfect AUPR with a decision boundary close to ground truth (Figure 3d).

**Neural networks**  Shallow methods cannot always be used (e.g. non-tabular or high-dimensional data), so we also experiment on NNs with hidden layers. Though straightforward, our equality separator and bump activation achieves better NN performance; equality separators and RBF separators with RBF activations consistently obtain perfect AUPR (Table 2), and we validate that closing numbers are useful in quantifying the ability to induce closed decision boundaries. Looking at individual models,

---

[4]This is according to neuron activation and differs from AD literature. More comments in Appendix F.3.

[5]Note that these are unsupervised methods. We use them only to benchmark and not as our main comparisons.

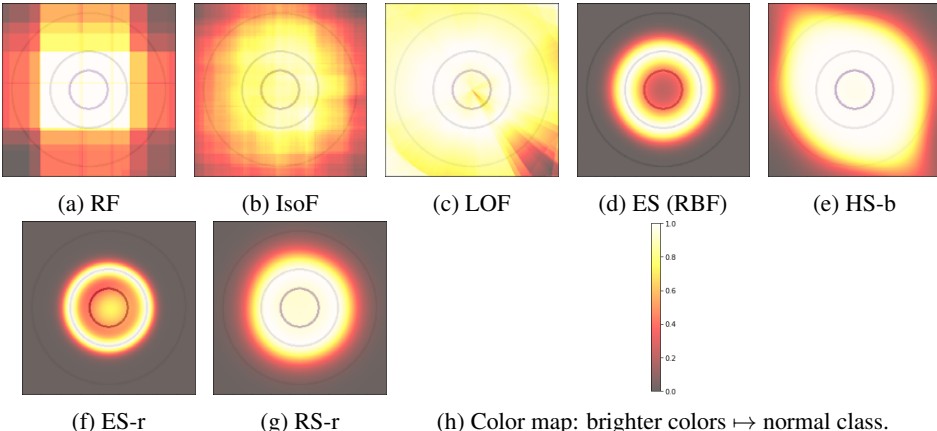

(a) RF      (b) IsoF      (c) LOF      (d) ES (RBF)      (e) HS-b

(f) ES-r      (g) RS-r      (h) Color map: brighter colors $\mapsto$ normal class.

Figure 3: Sample heatmap predictions by different models. The middle circle is the positive class, while the outer and inner circle is the negative class during training and testing respectively. Figures 3e, 3f and 3g are deep models, with hidden layer activations represented by the suffix 'b' for bump and 'r' for RBF. Shallow equality separator (Figure 3d) has a one-class decision boundary closest to ground truth, followed by equality separator neural networks with RBF activation (Figure 3f).

about half of the equality separators achieve high separation between normal and anomaly classes, with anomalies 8 times further from the hyperplane in the penultimate layer than normal data. In contrast, only 1 out of 20 RBF separators with RBF activations attain such high separation – most models produced outputs similar to Figure 3g. The semi-locality of equality separators retain locality for AD while enhancing separation between normal and anomaly classes of NNs with RBF activations.

## 4.2 SUPERVISED ANOMALY DETECTION: REAL-WORLD DATA

**Cyber-attacks** We use the NSL-KDD network intrusion dataset (Tavallaee et al., 2009) for real-world cyber-attack experiments. To simulate unseen attacks, training data only has normal data and denial of service attacks (DoS), and test data has DoS and unseen attacks, probe, privilege escalation and remote access attacks. For simplicity, we consider NNs with only 1 type of activation function in terms of locality as vanilla binary classifiers. Baselines we compare against are SVMs (binary classifiers) and OCSVM, IsoF and LOF (unsupervised AD). The state-of-the-art (SOTA) deep AD methods we compare against make use of labels: (1) negative sampling (NS), where binary classifiers are trained with generated anomalies (Sipple, 2020) and (2) SAD, where AD is based on hypersphere membership (Ruff et al., 2020). As a note, SAD is essentially an RBF separator on a pre-trained autoencoder. We report results in Table 3, comparing methods within their categories.

Due to their similarity, we expect some generalization of DoS to probe attacks, while little is expected for privilege escalation or remote access attacks. Our SVM baseline validates this, achieving better than random for probe attacks but worse for privilege and access attacks. Among unsupervised benchmarks, we see a trade-off between detecting known and unknown anomalies: OCSVM is best at detecting privilege and access attacks, while IsoF is best at detecting DoS and probe attacks. Notably, unsupervised methods are not competitive with the supervised SVM on seen (DoS) attacks.

Across the NN methods, we notice that equality separators/bump activations are the most competitive at detecting both seen and unseen anomalies, often obtaining the largest AUPR in their category (binary classifiers, NS and SAD). In particular, equality separation with SAD and standard binary classification seem to be among the best, significantly outperforming the SVM baseline on unseen attacks.

**Medical** We use the thyroid dataset (Quinlan, 1987) for medical anomalies. Using hyperfunction as the seen anomaly and subnormal as unseen, we compare ERM methods in Table 4a. Equality separation detects seen and unseen anomalies better than halfspace separation.

**Low data, high-dimensional images** MVTec dataset (Bergmann et al., 2019) has images of normal and defective objects. We select one anomaly class to be seen during training for supervised AD,

Table 3: Test AUPR on NSL-KDD. Negative sampling is denoted with a suffix (-NS), same for SAD. Overall AUPR against all attacks is also reported. Random classifiers are calculated in expectation.

| Model\Attack | DoS | Probe | Privilege | Access | Overall |
|---|---|---|---|---|---|
| Random | 0.435 | 0.200 | 0.007 | 0.220 | 0.569 |
| SVM | **0.950**±**0.000** | 0.626±0.000 | 0.003±0.000 | 0.123±0.000 | 0.894±0.000 |
| OCSVM | 0.779±0.000 | 0.827±0.000 | **0.405**±**0.000** | **0.760**±**0.000** | **0.897**±**0.000** |
| IsoF | 0.765±0.073 | **0.850**±**0.066** | 0.089±0.044 | 0.392±0.029 | 0.865±0.031 |
| LOF | 0.495±0.000 | 0.567±0.000 | 0.039±0.000 | 0.455±0.000 | 0.718±0.000 |
| HS | 0.944±0.016 | **0.739**±**0.018** | 0.016±0.006 | 0.180±0.033 | 0.877±0.010 |
| ES (ours) | **0.974**±**0.001** | 0.717±0.107 | **0.045**±**0.014** | **0.510**±**0.113** | **0.941**±**0.014** |
| RS | 0.356±0.002 | 0.213±0.002 | 0.010±0.000 | 0.228±0.002 | 0.406±0.001 |
| HS-NS | 0.936±0.001 | 0.642±0.030 | 0.006±0.000 | 0.139±0.001 | 0.845±0.006 |
| ES-NS (ours) | **0.945**±**0.009** | **0.659**±**0.013** | **0.023**±**0.011** | 0.206±0.013 | **0.881**±**0.007** |
| RS-NS | 0.350±0.002 | 0.207±0.002 | 0.009±0.000 | **0.223**±**0.002** | 0.401±0.002 |
| HS-SAD | 0.955±0.003 | 0.766±0.011 | **0.100**±**0.000** | 0.447±0.000 | 0.935±0.007 |
| ES-SAD (ours) | **0.960**±**0.002** | **0.795**±**0.004** | 0.047±0.002 | **0.509**±**0.009** | **0.952**±**0.002** |
| RS-SAD | 0.935±0.000 | 0.678±0.000 | 0.022±0.000 | 0.142±0.000 | 0.846±0.000 |

Table 4: Test AUPR on thyroid and MVTec dataset across anomalies.

(a) Thyroid AUPR.

| Diagnosis | HS | RS | ES (ours) |
|---|---|---|---|
| Hyper. | 0.907±0.013 | 0.030±0.002 | **0.934**±**0.004** |
| Subnorm. | 0.459±0.040 | 0.082±0.002 | **0.596**±**0.007** |
| Overall | 0.615±0.032 | 0.103±0.001 | **0.726**±**0.004** |

(b) MVTec overall AUPR.

| Object | HS | RS (SAD) | ES (ours) |
|---|---|---|---|
| Capsule | 0.918±0.002 | 0.894±0.000 | **0.947**±**0.003** |
| Hazelnut | **0.932**±**0.006** | 0.783±0.000 | **0.932**±**0.034** |
| Leather | **1.000**±**0.000** | 0.848±0.000 | 0.996±0.001 |
| Pill | 0.917±0.005 | 0.908±0.000 | **0.933**±**0.014** |
| Wood | 0.984±0.004 | 0.866±0.000 | **0.986**±**0.007** |
| Zipper | 0.998±0.000 | 0.879±0.000 | **1.000**±**0.000** |

(c) MVTec pills AUPR.

| Defect | HS | ES (ours) |
|---|---|---|
| Combine | 0.913±0.010 | **0.943**±**0.016** |
| Contam. | 0.768±0.029 | **0.802**±**0.114** |
| Crack | 0.652±0.009 | **0.730**±**0.046** |
| Imprint | 0.470±0.035 | **0.573**±**0.094** |
| Type | 0.885±0.071 | **0.910**±**0.156** |
| Scratch | 0.655±0.021 | **0.690**±**0.058** |

while the other anomaly classes are only seen during test time. There are on the order of 300 training samples, while feature embeddings (from DINOv2 (Oquab et al., 2023)) is 1024-dimensional.

We compare AUPR of binary classifiers across 6 objects in Table 4b, zooming in on individual defect detection performance for pills in Table 4c. Since DINOv2 acts as a kernel to embed features into a high dimensional space, we consider halfspace separation as our baseline, while RBF separators act like SAD in this case for comparison. During training, we observe that RBF separators consistently have high training error and low train AUPR (below 0.55), suggesting underfitting. Here, too much locality impedes learning, especially in high dimensions. Meanwhile, we see equality separators detecting unseen anomalies the best for 5/6 of the objects. Furthermore, equality separators are consistently the best across different anomaly types (Table 4c).

**Limitations**   We corroborate with Flake (1993) that bump activations in equality separation can be sensitive to initializations, evidenced by some variance in test AUPR. To mitigate this, we can set $\sigma$ to be large to mitigate this. Ensembling equality separators can also help suppress inaccurate outputs from bad initializations. We discuss more in Appendix F.3. We also note that NNs can struggle at generalizing to unseen data, as OCSVM achieves better AUPR than NNs for the unseen attacks in NSL-KDD. We posit that understanding how the activation function, loss function and optimization scheme work together in equality separation is the next step towards designing better NNs for AD.

## 5   CONCLUSION

We revisit the belief that XOR is not linearly separable, which prompts us to propose a new binary classification paradigm, *equality separation*. Equality separators use the distance of a datum from a hyperplane as their classification metric. They have twice the VC dimension of halfspace separators (e.g. it can solve some linearly nonseparable problems like XOR) and enjoy the benefits of globality and locality, which we formalize and quantify with *closing numbers*. Using the $\epsilon$-error separator and its smooth approximation, the *bump activation*, we can integrate equality separation with neural networks to form closed decision regions for anomaly detection. We support these claims empirically, quantitatively and qualitatively showing that equality separation can produce better one-class decision boundaries, striking a good balance detecting seen and unseen anomalies in both low and high dimensions.

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

## APPENDIX A   MORE CONNECTIONS OF EQUALITY SEPARATORS

We explore more connections of equality separators with other concepts in machine learning.

### A.1   CONNECTION TO HYPERRECTANGLE CLASSIFIERS

Since the halfspace separator is tightly connected to our decision rule, our decision rule is also connected to other classifiers that are related to halfspace separators. To illustrate this, we use one example of the hypothesis class being the set of all hyperrectangles in $\mathbb{R}^n$. Just as how using hyperrectangles as classifiers in $\mathbb{R}^n$ can be seen as using the intersection of $2n$ halfspaces, we can also see its relation to our $\epsilon$-error separator.

For each dimension $i \in [n]$, define a weight vector $\mathbf{w}_i \in \mathbb{R}^n$, a bias term $b_i \in \mathbb{R}$ and a width parameter $\epsilon_i \in \mathbb{R}^n$ that will control the width of the hyperrectangle in the $i^{th}$ dimension. Hyperrectangle classifiers can be produced by the intersection of these $n$ $\epsilon$-error separators. In fact, we can also generalize this to hyperparallelograms too, by removing the constraint of the orthogonality of the weight vector.

### A.2   CONNECTION WITH LINEAR AND SUPPORT VECTOR REGRESSION

Due to the equality form of $\mathbf{w}^T\mathbf{x} + b = 0$, our $\epsilon$-error separator can be seen as a linear regression problem for one class, with $\epsilon$ defining the error/noise tolerated. This is similar to the formulation of support vector regression, where the goal of regression is such that points within the margin are not penalized for being a small distance away from the hyperplane.

Our approach differs in how we handle anomalies. Our optimization scheme accounts for information provided by the other class, which can act as a form of regularization. In a vanilla linear regression problem, the negative class may not matter so much during training, and we can discard the negative class during training and perform linear regression with only the positive class. However, in a learning setting with a non-linear embedding function (like a neural network), the negative class can help with updating the embedding function such that the negative class will not lie on the hyperplane (and can help prevent degenerate solutions, such as projecting all points onto a single point). For a classification task, the distance from the hyperplane determines how similar a data point is to the class defined by the hyperplane. This inductive bias is especially useful for one-class classification.

## APPENDIX B   VC DIMENSION PROOFS AND COMMENTS

As a progression, we will first prove the VC dimension in $\mathbb{R}^2$ of the strict equality separator for binary variables, then for the whole space of $\mathbb{R}^2$ and finally, the general version of $\mathbb{R}^n$. The former two will provide good intuitions for the generalization to $\mathbb{R}^n$. We note that we will assume that $n \geq 2$ in this section since the case for $n = 1$ is trivial. We denote $\mathcal{H}$ as the hypothesis class of strict equality separators, as defined in Definition 2.1.

### B.1   BINARY (BOOLEAN) VARIABLES IN $\mathbb{R}^2$

Let $VCdim(\mathcal{F})$ denote the VC-dimension of a function class $\mathcal{F}$. In the special case where $\mathbf{x}$ is binary (ie. has entries which are binary variables) in $\mathbb{R}^2$, we can observe that $VCdim(\mathcal{H}) = 4$, which is the maximum number of points in the binary feature space in $\mathbb{R}^2$.

To see how one can always shatter the 4 points of $(0,0), (0,1), (1,0)$ and $(1,1)$, we can use the following rule:

1. If there is a class that does not have any point that belongs to it, then a line passing through none of the points suffices as a classifier.

2. If there is a class that has only 1 point that belongs to it, then a line passing through that point that does not pass through any other point suffices as a classifier. Examples are given in the `AND` and `OR` cases.

3. If there is a class that has only 2 points that belong to it, then a line passing through those 2 points as a classifier. Example is given in the `XOR` cases.

Other cases are equivalent to label flipping, which can be captured by the hypothesis class $\mathcal{H}$. Hence, the set of 4 points can be shattered by $\mathcal{H}$.

### B.2   VC DIMENSION IN $\mathbb{R}^2$

In fact, we can analyse a more general setting where the elements of $\mathbf{x}$ live in $\mathbb{R}^2$ rather than being contrained to binary variables. In this more general setting, we obtain the following theorem.

**Theorem B.1.** *Let* $\mathcal{H} = \{\mathbb{I}(\mathbf{w}^T\mathbf{x}+b = 0)\} \cup \{\mathbb{I}(\mathbf{w}^T\mathbf{x}+b \neq 0)\}$ *and* $\mathbf{x} \in \mathbb{R}^2$. *Then,* $VCdim(\mathcal{H}) = 5$.

The VC-dimension of $\mathcal{H}$ in $\mathbb{R}^2$ is an increase of 2 from the VC-dimension of the half-space separator. The increase in 2 is due to the assignment rule as proposed in Appendix B.1.

To prove the VC-dimension, we will construct a 2-sided inequality with lemmas B.2 and B.3.

**Lemma B.2.** *Let* $\mathcal{H} = \{\mathbb{I}(\mathbf{w}^T\mathbf{x}+b = 0)\} \cup \{\mathbb{I}(\mathbf{w}^T\mathbf{x}+b \neq 0)\}$ *and* $\mathbf{x} \in \mathbb{R}^2$. *Then,* $VCdim(\mathcal{H}) \geq 5$.

*Proof.* We find a set of 5 distinct points that are shattered by $\mathcal{H}$. Let these 5 points be arranged on a unit circle centred at the origin, equally spaced (in the complex plane, the points will be the $5^{th}$ roots of unity). Following the assignment rule in Appendix B.1, we can always draw a line that classifies the points. Therefore, these 5 points are shattered by $\mathcal{H}$.   $\square$

**Lemma B.3.** *Let* $\mathcal{H} = \{\mathbb{I}(\mathbf{w}^T\mathbf{x}+b = 0)\} \cup \{\mathbb{I}(\mathbf{w}^T\mathbf{x}+b \neq 0)\}$ *and* $\mathbf{x} \in \mathbb{R}^2$. *Then,* $VCdim(\mathcal{H}) \leq 5$.

*Proof.* We will prove that any 6 distinct points cannot be shattered by $\mathcal{H}$.[6] Consider 2 exhaustive cases: (i) when there does not exist 3 colinear points and (ii) when there exists 3 points that are colinear (ie. lie on the same line) .

In case (i), any labelling that has 3 points from class 0 and the remaining 3 points from class 1 will not produce a perfect classifier. Since 3 points belong to each class, then if there exists a perfect classifier from $\mathcal{H}$, then the line must pass through all 3 points from either class. Since no 3 points are colinear, then there is no line that passes through 3 points from either class. Therefore, any 6 points that do not have 3 colinear points are not shattered by $\mathcal{H}$.

---

[6]The case where points are not distinct is trivial to show the inability to shatter, so we will not explicitly mention this fact in future analyses.

In case (ii), it is not possible to shatter the 6 points. Denote the concept (ie. the true function) as $c : \mathcal{X} \to \mathcal{Y}$. Consider 3 colinear points $\mathbf{a}_1, \mathbf{a}_2, \mathbf{a}_3 \in \mathbb{R}^2$. Denote the other points as $\mathbf{d}_1, \mathbf{d}_2, \mathbf{e} \in \mathbb{R}^2$. Let $c(\mathbf{a}_1) = c(\mathbf{d}_1) = c(\mathbf{d}_2) = 0$ and $c(\mathbf{a}_2) = c(\mathbf{a}_3) = c(\mathbf{e}) = 1$. The linear classifier from $\mathcal{H}$ has to be a line that lies on all the classes of 1 class. However, there is no line that lies on all the points from class 1, because that line will either

1. contain a point from class 0 (e.g. the line passing through $\mathbf{a}_2$ and $\mathbf{a}_2$ also passes through $\mathbf{a}_1$), or

2. not contain all the points from class 1 (because a line that does not pass through $\mathbf{a}_2$ and $\mathbf{a}_3$ will not have $\mathbf{a}_2$ or $\mathbf{a}_3$).

Hence, the line needs to pass through all 3 points from class 0. If such a line does not exist, then the proof is done. If such a line exists, then there exists some points $\mathbf{d}_1, \mathbf{d}_2 \in \mathbb{R}^2$ that are colinear to $\mathbf{a}_1$ but not to $\mathbf{a}_2, \mathbf{a}_3$ or $\mathbf{e}$. Then, define a new labelling function $c'$ that switches the labels of $\mathbf{d}_2$ and $\mathbf{e}$. In other words, $c'(\mathbf{a}_1) = c'(\mathbf{d}_1) = c'(\mathbf{e}) = 0$ and $c'(\mathbf{a}_2) = c'(\mathbf{a}_3) = c'(\mathbf{d}_2) = 1$. Then, we face a similar difficulty in classifying the colinear points $\mathbf{a}_1, \mathbf{d}_1$ and $\mathbf{e}$, because no line contains all class 0 points. Since there exists a labelling where no hypothesis from $\mathcal{H}$ can classify these 6 points, then any 6 points that have 3 colinear points are not shattered by $\mathcal{H}$.

Thus, there does not exist any set of 6 points that can be shattered by $\mathcal{H}$. $\qquad\square$

### B.3  VC DIMENSION IN $\mathbb{R}^n$

**Theorem B.4.** *Let $\mathbf{x} \in \mathbb{R}^n$ for $n \in \{2, 3, ...\}$, and the hypothesis class be the strict equality separator $\mathcal{H} = \{\mathbb{I}(\mathbf{w}^T \mathbf{x} + b = 0)\} \cup \{\mathbb{I}(\mathbf{w}^T \mathbf{x} + b \neq 0)\}$. Then, $VCdim(\mathcal{H}) = 2n + 1$.*

The classical halfspace separator gives a VC dimension of $n + 1$. An intuitive way to interpret the factor of 2 in the $n$-term is that we allow 2 kinds of hypotheses – one that asserts equality and one that does not. In essence, our hypothesis class allows for label flipping, which gives it the flexibility of expression. Hence, the equality separator hypothesis class is on the order of twice as expressive as the halfspace separator. However, allowing label flipping for the halfspace separator does not increase its expressivity. The asymmetry of the equality separator gives rise to this property that label flipping is indeed useful, doubling its expressivity.

We break the claim into a 2-sided inequality with lemmas B.5 and B.6.

**Lemma B.5.** *Let $\mathcal{H} = \{\mathbb{I}(\mathbf{w}^T \mathbf{x} + b = 0)\} \cup \{\mathbb{I}(\mathbf{w}^T \mathbf{x} + b \neq 0)\}$ and $\mathbf{x} \in \mathbb{R}^n$. Then,*

$$VCdim(\mathcal{H}) \geq 2n + 1.$$

*Proof.* Consider a set of $2n + 1$ distinct points such that for any hyperplane in $\mathbb{R}^n$, only a maximum of $n$ points lie on the hyperplane. Then, for any labelling of the $2n + 1$ points, let the positive class be the class that has the fewest data points that belong to it. (We can do this because our hypothesis class allows for label flipping.) Let the number of data points belonging to this positive class be $k$. Notice that $k \leq \lfloor \frac{2n+1}{2} \rfloor = n$. Denote the non-empty set of all hyperplanes that contain all of these at most $k$ points as $H$.

If $k = n$, then $|H| = 1$ (ie. the hyperplane is unique). Note that this unique hyperplane does not contain any other points, by construction of the data points, so the hyperplane is a perfect classifier on this labelling.

Otherwise, if $k < n$, then $|H| = \infty$ (ie. there are infinitely many hyperplanes). With induction, we prove that there is a hyperplane that passes through $k$ points that does not pass through the other $(2n + 1) - k$ points.

**Base case:** $k = n - 1$.   For any set of $k$ points, let $H$ be the set of all hyperplanes that pass through these $k$ points, and let $H' \subseteq H$ be the set of all hyperplanes that pass through the $k$ points but also pass through at least 1 of the other $(2n+1) - k$ points. Then, $|H| = \infty$ while $|H'| = (2n+1) - k = n+2$ because the other $n + 2$ points (ie. the points from the negative class) restrict us from using those hyperplanes for classification. Since $|H| > |H'|$ and $H' \subseteq H$, then $H'$ is a proper subset of $H$, and

there exists a hyperplane in $H$ that is not in $H'$. By definition, this hyperplane contains all $k$ points and none of the other $(2n + 1) - k$ points.

**Inductive step: Assume that there exists a hyperplane that passes through any $k + 1$ points and does not pass through the other $(2n + 1) - (k + 1)$ points, for $0 \leq k < n - 1$. Then, we prove that there exists a hyperplane that passes through any $k$ points.** Consider any set of $k + 1$ points (and denote this as $P$) and consider a hyperplane $h$ that passes through the $k + 1$ points but not the other $2n - k$ points. Let the set of the other $2n - k$ points be $Q$ (ie. $P$ and $Q$ form a partition of the set of all $2n + 1$ points). Let $\mathbf{p} \in P$ be any point in $P$ (since $P$ is non-empty). Then, there is an added free variable (ie. added degree of freedom) to define a hyperplane that passes through $P\backslash\{\mathbf{p}\}$. To avoid passing through points in $Q \cup \{\mathbf{p}\}$, this free variable is unable to take on a finite number of values to avoid the $(2n + 1) - k$ points. Since the variable can take on an infinite number of values except a finite number of values, there is a value that this variable can take on. Using this value, we can find a hyperplane that passes through all $k$ points in $P$ while not passing through the $(2n + 1) - k$ points in $Q \cup \{\mathbf{p}\}$.

By induction, we have shown that for all $k \in \{0, 1, ..., n - 1\}$, that there exists a hyperplane that passes through $k$ points but not the other $(2n + 1) - k$ points. Hence, there always exists a hyperplane to classify the $k$ points from the positive class such that none of the $(2n + 1) - k$ points from the negative class are on this hyperplane.

Since a perfect classifier from $\mathcal{H}$ always exists for the aforementioned dataset for any labelling, then there exists a set of $2n + 1$ points in $\mathbb{R}^n$ that $\mathcal{H}$ can shatter. Thus, $VCdim(\mathcal{H}) \geq 2n + 1$. $\qquad\square$

**Lemma B.6.** *Let $\mathcal{H} = \{\mathbb{I}(\mathbf{w}^T\mathbf{x} + b = 0)\} \cup \{\mathbb{I}(\mathbf{w}^T\mathbf{x} + b \neq 0)\}$ and $\mathbf{x} \in \mathbb{R}^n$. Then,*

$$VCdim(\mathcal{H}) \leq 2n + 1.$$

The proof only requires basic linear algebra and the idea is as follows. We desire to find a labeling on $2n + 2$ points such that there is no strict equality separator for that labeling. For some labeling, we consider (translated) subspaces of the 2 classes defined by the points from that class and relabel points until we find a labeling that satisfies the above condition. In the arguments, we often consider 2 classes with $n$ points each, and leave the remaining 2 points to help us produce the desired labeling by labeling the 2 points with the corresponding labels and sometimes swapping the labeling of points already with labels.

*Proof.* Consider any set of distinct $2n + 2$ points in $\mathbb{R}^n$, and denote this set as $X$. We aim to show that there exists a labelling on these points such that no $h \in \mathcal{H}$ can classify the points perfectly (ie. $\mathcal{H}$ cannot shatter any set of $2n + 2$ points). We will exhaustively consider cases and show that in all cases, there exists a labelling such that no classifier in $\mathcal{H}$ can perfectly classify all the points on such a labelling.

Note that if $X$ lies on a hyperplane, then there exists a labelling half of $X$ as positive and the other half as negative that would fail to produce a perfect classifier. Hence, we consider the fact that $X$ does not lie on one hyperplane.

Let $\{P_1, P_2, Q\}$ be a partition of $X$, where $|P_1| = |P_2| = n$ and $|Q| = 2$. Let $M : A \rightarrow \mathbb{R}^{n \times |A|}$ be the function that outputs a matrix where each column is an element of $A$. Formally, for non-empty, finite set $A := \{\mathbf{a}_1, ..., \mathbf{a}_{|A|}\} \subseteq X$, define $M$ as a function that generates the matrix $M(A) = [\mathbf{a}_1, ..., \mathbf{a}_{|A|}]$ from $A$.

**Case 1** If there exists a partition of $X$ such that $M(P_1)$ and $M(P_2)$ are not full rank, then consider points $\mathbf{b} \in P_1$ and $\mathbf{c} \in P_2$ such that $\mathrm{rank}(M(P_1\backslash\{\mathbf{b}\})) = \mathrm{rank}(M(P_1))$ and $\mathrm{rank}(M(P_2\backslash\{\mathbf{c}\})) = \mathrm{rank}(M(P_2))$. Such points exist based on a counting argument, given that $M(P_1)$ and $M(P_2)$ are not full rank. Then, consider the labelling such that $P_1\backslash\{\mathbf{b}\}, \mathbf{c}$ belong to class one and $P_2\backslash\{\mathbf{c}\}, \mathbf{b}, Q$ belong to class two. A perfect classifier from $\mathcal{H}$ does not exist on this labelling: using a hyperplane generated by $P_1\backslash\{\mathbf{b}\}$[7] to label class one as the positive class will classify $\mathbf{b}$ wrongly because $\mathbf{b} \in \mathrm{col}(M(P_1\backslash\{\mathbf{b}\}))$ (by assumption) will be classified as negative although it is from the positive

---

[7]A hyperplane such that all points in $P_1\backslash\{\mathbf{b}\}$ lie on it.

class. Likewise, using a hyperplane generated by $P_2 \backslash \{\mathbf{c}\}$ to label class two as the positive class will classify $\mathbf{c}$ wrongly because $\mathbf{c} \in \text{col}(M(P_2 \backslash \{\mathbf{c}\}))$.

Note that using hyperplanes other than the ones described will also fail to produce perfect classifiers, because points in $P_1$ (when class one is the positive class) or $P_2$ (when class two is the positive class) will not be classified correctly. We omit this comment in further analyses.

**Case 2**  Otherwise, consider the case where there exists a partition of $X$ such that either $M(P_1)$ or $M(P_2)$ is full rank. Without loss of generality, let $M(P_2)$ be rank deficient, while $M(P_1)$ is full rank. By the same counting argument, there exists non-empty set $K \subseteq P_2$ such that $\text{rank}(M(P_2 \backslash K)) = \text{rank}(M(P_2))$.

If there is a point $\mathbf{p} \in K$ for any $K$ such that $\mathbf{p} \notin \text{col}(M(P_1))$, then consider the labelling such that $P_1, p$ belong to class one and $P_2 \backslash \{\mathbf{p}\}, Q$ belong to class two. A perfect classifier from $\mathcal{H}$ does not exist on this labelling: using the unique hyperplane generated by $P_1$ to label class one as the positive class will classify $\mathbf{p}$ wrongly, because $\mathbf{p} \notin \text{col}(M(P_1))$. On the other hand, using a hyperplane generated by $P_2 \backslash \{\mathbf{p}\}$ to label class two as the positive class will classify $\mathbf{p}$ wrongly because $\mathbf{p} \in \text{col}(M(P_2 \backslash \{\mathbf{p}\}))$.

Thus, consider that there is no such $\mathbf{p}$ that exists (ie. for any $K$ that fulfills the constraints, $\mathbf{p} \in K$ will belong to the column space of $M(P_1)$). Then, pick $K$ such that $K$ is the largest possible set that fulfills the constraints (ie. $\text{rank}(M(P_2 \backslash K)) = \text{rank}(M(P_2))$). Note that $|K| \geq 1$.

If $|K| > 1$, then consider 2 distinct points $\mathbf{p}, \mathbf{q} \in K$ and the labelling such that $P_1, \mathbf{p}, Q$ belong to class one and $P_2, \backslash \{\mathbf{p}\}$ A perfect classifier from $\mathcal{H}$ does not exist on this labelling: using the unique hyperplane generated by $P_1$ to label class one as the positive class will classify $\mathbf{q}$ wrongly, because $\mathbf{q} \in \text{col}(M(P_1))$. On the other hand, using a hyperplane generated by $P_2 \backslash \{\mathbf{p}\}$ to label class two as the positive class will classify $\mathbf{p}$ wrongly because $\mathbf{p} \in \text{col}(M(P_2 \backslash \{\mathbf{p}\}))$.

Otherwise, if $|K| = 1$, then consider $\mathbf{x} \in P_1$ such that $P_1 \backslash \{\mathbf{x}\} \cup K$ generates a unique hyperplane which is the same hyperplane as the unique hyperplane generated by $P_1$. Such a point exists because $\mathbf{p} \in K$ is a linear combination of points in $P_1$. Hence, such a point $\mathbf{x}$ exists.[8] Then, consider the labelling where $P_1 \backslash \{\mathbf{x}\}, K$ belong to class one and $P_2 \backslash K, \mathbf{x}, Q$ belong to class two. A perfect classifier from $\mathcal{H}$ does not exist on this labelling: using the unique hyperplane generated by $P_1 \backslash \{\mathbf{x}\} \cup K$ to label class one as the positive class will classify $\mathbf{x}$ wrongly, because $\mathbf{x} \in \text{col}(M(P_1))$. On the other hand, using a hyperplane generated by $P_2 \backslash K$ to label class two as the positive class will classify $\mathbf{p} \in K$ wrongly because $\mathbf{p} \in \text{col}(M(P_2 \backslash \{\mathbf{p}\}))$.

**Case 3**  Lastly, we consider the case such that there is no such partition such that both $M(P_1)$ and $M(P_2)$ are not rank deficient. In other words, the 2 hyperplanes generated by $P_1$ and $P_2$ respectively are both unique. Consider $Q := \{\mathbf{p}, \mathbf{q}\}$.

If both $\mathbf{p}$ and $\mathbf{q}$ do not lie in the column space of either $M(P_1)$ or $M(P_2)$, then consider the labelling where $P_1, \mathbf{p}$ belong to class one and $P_2, \mathbf{q}$ belong to class two. A perfect classifier from $\mathcal{H}$ does not exist on this labelling: using the unique hyperplane generated by $P_1$ to label class one as the positive class will classify $\mathbf{q}$ wrongly, because $\mathbf{p} \notin \text{col}(M(P_1))$. On the other hand, using the unique hyperplane generated by $P_2$ to label class two as the positive class will classify $\mathbf{q}$ wrongly because $\mathbf{q} \notin \text{col}(M(P_2))$.

Otherwise, there exists a point in $Q$ that lies in the column space of either $M(P_1)$ or $M(P_2)$. Consider the case where one point does not lie in the column space of either $M(P_1)$ or $M(P_2)$. Without loss of generality, let $\mathbf{p}$ be the point that does not lie in the column space of either $M(P_1)$ or $M(P_2)$, and let $\mathbf{q} \in \text{col}(M(P_2))$. If $\text{col}(M(P_2)) \subseteq \text{col}(M(P_1))$, then consider the labelling where $P_1, \mathbf{q}$ belong to class one and $P_2, \mathbf{p}$ belong to class two. A perfect classifier from $\mathcal{H}$ does not exist on this labelling: using the unique hyperplane generated by $P_1$ to label class one as the positive class will classify some point $\mathbf{x} \in P_2$ wrongly, because $\mathbf{x} \in \text{col}(M(P_2)) \subseteq \text{col}(M(P_1))$ is classified as the positive class although it belongs to the negative class. Meanwhile, using the unique hyperplane generated by $P_2$ to label class two as the positive class will classify $\mathbf{p}$ wrongly because $\mathbf{p} \notin \text{col}(M(P_2))$.
Else, $\text{col}(M(P_2)) \not\subseteq \text{col}(M(P_1))$, so there exists a point $\mathbf{s} \in P_2$ such that $\mathbf{s} \notin \text{col}(M(P_1))$. Then, $M(P_2 \backslash \{\mathbf{s}\} \cup \{\mathbf{q}\})$ is full rank because $\mathbf{q}$ is a linear combination of all points in $P_2$. Note that the

---

[8]We repeat a similar argument in future analyses, and omit this comment in the future for brevity.

unique hyperplane generated by $P_2\backslash\{\mathbf{s}\} \cup \{\mathbf{q}\}$ is the same hyperplane as the unique hyperplane generated by $P_2$. Then consider the labelling where $P_1, \mathbf{s}$ belong to class one and $P_2\backslash\{s\}, \mathbf{q}, \mathbf{p}$ belong to class two. A perfect classifier from $\mathcal{H}$ does not exist on this labelling: using the unique hyperplane generated by $P_1$ to label class one as the positive class will classify $\mathbf{s}$ wrongly, because $\mathbf{s} \notin \mathrm{col}(M(P_1))$. On the other hand, using the unique hyperplane generated by $P_2\backslash\{\mathbf{s}\} \cup \{\mathbf{q}\}$ to label class two as the positive class will classify $\mathbf{p}$ wrongly because $\mathbf{p} \notin \mathrm{col}(M(P_2))$.

Hence, consider the case where both $\mathbf{p}$ and $\mathbf{q}$ lie on the column space of $M(P_1)$ or $M(P_2)$. If they lie on different column spaces, without loss of generality, let $\mathbf{p} \in \mathrm{col}(M(P_1))$ and $\mathbf{q} \in \mathrm{col}(M(P_2))$. Note that $\mathbf{p} \notin \mathrm{col}(M(P_2))$ and $\mathbf{q} \notin \mathrm{col}(M(P_1))$ by our assumption. Then consider the labelling where $P_1, \mathbf{q}$ belong to class one and $P_2, \mathbf{p}$ belong to class two. A perfect classifier from $\mathcal{H}$ does not exist on this labelling: using the unique hyperplane generated by $P_1$ to label class one as the positive class will classify $\mathbf{q}$ wrongly, because $\mathbf{q} \notin \mathrm{col}(M(P_1))$. On the other hand, using the unique hyperplane generated by $P_2$ to label class two as the positive class will classify $\mathbf{p}$ wrongly because $\mathbf{p} \notin \mathrm{col}(M(P_2))$.

Thus, we consider the case where $\mathbf{p}$ and $\mathbf{q}$ lie on the same column space. Without loss of generality, let $\mathbf{p}, \mathbf{q} \in \mathrm{col}(M(P_1))$. If there is a point that does not lie in the column space of $M(P_2)$, then without loss of generality, let this point be $\mathbf{p}$. Consider a point $\mathbf{x} \in P_1$ the unique hyperplane generated by $P_1\backslash\{\mathbf{x}\} \cup \{\mathbf{q}\}$ is the same hyperplane as the unique hyperplane generated by $P_1$, because $\mathbf{q}$ is a linear combination of points in $P_1$. Then consider the labelling where $P_1\backslash\{\mathbf{x}\}, \mathbf{q}$ belong to class one and $P_2, \mathbf{p}, \mathbf{x}$ belong to class two. A perfect classifier from $\mathcal{H}$ does not exist on this labelling: using the unique hyperplane generated by $P_1$ to label class one as the positive class will classify $\mathbf{x}$ wrongly, because $\mathbf{x} \in \mathrm{col}(M(P_1))$ will be labelled as positive although it is from the negative class. On the other hand, using the unique hyperplane generated by $P_2$ to label class two as the positive class will classify $\mathbf{p}$ wrongly because $p \notin \mathrm{col}(M(P_2))$.

Otherwise, consider the case where both $\mathbf{p}, \mathbf{q} \in \mathrm{col}(M(P_1))$ and $\mathbf{p}, \mathbf{q} \in \mathrm{col}(M(P_2))$. Then consider the labelling where $P_1, \mathbf{q}$ belong to class one and $P_2, \mathbf{p}$ belong to class two. A perfect classifier from $\mathcal{H}$ does not exist on this labelling: using the unique hyperplane generated by $P_1$ to label class one as the positive class will classify $\mathbf{p}$ wrongly, because $\mathbf{p} \in \mathrm{col}(M(P_1))$ will be labelled as positive although it is from the negative class. On the other hand, using the unique hyperplane generated by $P_2$ to label class two as the positive class will classify $\mathbf{q}$ wrongly because $\mathbf{q} \in \mathrm{col}(M(P_2))$ will be labelled as positive although it is from the negative class.

We have exhaustively considered cases and showed that, for any set of $2n + 2$ distinct points, $\mathcal{H}$ is unable to shatter it. Thus, $VCdim(\mathcal{H}) \leq 2n + 1$. $\qquad\square$

### B.4 VC DIMENSION OF $\epsilon$-ERROR SEPARATOR

The lower bound of $2n + 1$ follows from Theorem 2.3, since the margin of the $\epsilon$-error separator can be made arbitrarily small.

The upper bound is from the union property of hypothesis classes found in Exercise 6.11 in Chapter 6.8 of Shalev-Shwartz & Ben-David (2014). The VC dimension of hyper-hill (equivalent to hyper-ridge) classifiers are $n + 1$ (Flake, 1993), so we have an upper bound of $(n + 1) + (n + 1) + 1 = 2n + 3$.

Work can be done to show that the VC dimension in $\mathbb{R}^2$ is $2n + 2$. We conjecture that this is true for all $n > 2$ as well, but leave this to future work. Note that for $n = 1$, the VC dimension is $2n + 1 = 3$ – the reduction in expressivity seems to arise from the distance of a given datum to be a given point rather than a line, plane or hyperplane (which extends to infinity, unlike a point).

### B.5 COMMENTS ON THE DELTA RULE

Faster convergence for NNs with bump activations has been attributed to a regularization term in a modified delta rule by Dawson & Schopflocher (1992) to encourage outputs to be on the hyperplane for the positive class. On the other hand, we provide an alternative heuristic by understanding the bump activation as a smooth version of the equality separator and motivate the learning theory parallel for this phenomenon – equality separators have a higher VC dimension than halfspace separators. Hence, faster convergence may be due to the increased complexity of equality separators that are encoded in the bump activation.

## APPENDIX C  PROOF FOR SEPARATING HYPERPLANE IN PSEUDO LIKELIHOOD RATIO TEST

We provide the proof in Section 2.2 that, when we use 2 parallel equality separators ($\mathbf{w}_1 = \mathbf{w}_2$) and equal class weighting ($t = 1$, ie. a false positive is as bad as a false negative) for binary classification, our pseudo likelihood ratio test

$$\mathbb{I}\left(\frac{\epsilon_2}{\epsilon_1} \geq t\right) = \mathbb{I}\left(\frac{1}{||\mathbf{w}_2||}|\mathbf{w}_2^T\mathbf{x}' + b_2| - \frac{t}{||\mathbf{w}_1||}|\mathbf{w}_1^T\mathbf{x}' + b_1| \geq 0\right)$$

decomposes to

$$\mathbb{I}((b_2 - b_1)(2\mathbf{w}_1^T\mathbf{x} + b_1 + b_2) \geq 0)$$

which corresponds to the separating hyperplane

$$2\mathbf{w}_1^T\mathbf{x} + b_1 + b_2.$$

*Proof.* Instead of dealing with absolute values, we take the square on both sides (which does not change the inequality because both sides are positive). With parallel hyperplane and equal class weighting, we are able to cancel out the square terms to obtain a separating hyperplane. The exact steps are detailed as follows:

$$
\begin{aligned}
\mathbb{I}\left(\frac{\epsilon_2}{\epsilon_1} \geq t\right) &= \mathbb{I}\left(\frac{\epsilon_2^2}{\epsilon_1^2} \geq t^2\right) \\
&= \mathbb{I}\left(\epsilon_2^2 \geq t^2\epsilon_1^2\right) \\
&= \mathbb{I}\left((\mathbf{w}_2^T\mathbf{x} + b_2)^2 \geq t^2(\mathbf{w}_1^T\mathbf{x} + b_1)^2\right) \\
&= \mathbb{I}\left((\mathbf{w}_2^T\mathbf{x} + b_2)^2 - t^2(\mathbf{w}_1^T\mathbf{x} + b_1)^2 \geq 0\right) \\
&= \mathbb{I}\left((\mathbf{w}_2^T\mathbf{x} + b_2)^2 - (\mathbf{w}_1^T\mathbf{x} + b_1)^2 \geq 0\right) & t = 1 \\
&= \mathbb{I}\left((\mathbf{w}_1^T\mathbf{x} + b_2)^2 - (\mathbf{w}_1^T\mathbf{x} + b_1)^2 \geq 0\right) & \mathbf{w}_1 = \mathbf{w}_2 \\
&= \mathbb{I}\left(2\mathbf{w}_1^T\mathbf{x}(b_2 - b_1) + (b_2^2 - b_1^2) \geq 0\right) & \text{expand the square} \\
&= \mathbb{I}\left((b_2 - b_1)(2\mathbf{w}_1^T\mathbf{x} + (b_2 + b_1)) \geq 0\right) \\
&= \mathbb{I}\left((b_2 - b_1)(2\mathbf{w}_1^T\mathbf{x} + b_1 + b_2) \geq 0\right)
\end{aligned}
$$

If $b_1 \neq b_2$, then we have

$$
\begin{aligned}
\mathbb{I}\left(\frac{\epsilon_2}{\epsilon_1} \geq t\right) &= \mathbb{I}\left((b_2 - b_1)(2\mathbf{w}_1^T\mathbf{x} + b_1 + b_2) \geq 0\right) \\
&= \mathbb{I}\left(2\mathbf{w}_1^T\mathbf{x} + b_1 + b_2 \geq 0\right).
\end{aligned}
$$

$\square$

We note that in the case of XOR, $b_1 = b_2$ and the hyperplane to classify each class is the same. Hence, there is no separating hyperplane between the 2 classes. This is another way we can view the linear inseparability of XOR.

## APPENDIX D    CLOSING NUMBERS

We first note that whenever we refer to a closed decision region in this section, we refer to a region with positive (i.e. non-zero) volume unless otherwise stated.

To prove that the closing number is $k$, it is helpful to show the existence of a closed decision region using the intersection of $k$ hypotheses and the impossibility using fewer (i.e. intersection of $k-1$ hypotheses).

### D.1    PROOF OF CLOSING NUMBER OF EQUALITY SEPARATORS IN $\mathbb{R}^n$

To show that the closing number is at most $n$ in $\mathbb{R}^n$, we observe that a hyperrectangle can be formed through the intersection of $n$ equality separators with orthogonal normal (ie. weight) vectors where the volume in the margin is considered the positive class. Since the volume of a hyperrectangle is finite, so is this intersection.

As an aside, as long as these normal vectors form a set of linearly independent vectors (more precisely, that they form a basis for $\mathbb{R}^n$), the finite-volume argument holds by observing a hyperparallelogram is formed.

If we have the intersection of $k < n$ equality separators, we observe a decision region with infinite volume or no volume. The first case is when one equality separator assigns the positive class to the region outside of the margin. This induces an infinite-volumed decision region for non-parallel normal vectored equality separators. Otherwise, the decision region either has no volume (when the intersection of parallel equality separators is a pair of hyperplanes, a hyperplane or the empty set) or infinite volume (in the more general case). The other case is where all equality separators assign a positive classification to the region within the margin. Without loss of generality, let the bias terms of all $k$ equality separators be 0 and consider the subspace $\mathcal{S}$ spanned by the $k$ normal vectors. Since $k < n$ (or, similarly, in the case where the set of weight vectors of equality separators do not form a basis for $\mathbb{R}^n$), then there exists a vector $\mathbf{v}$ that is perpendicular to subspace $\mathcal{S}$. The volume of the intersection $I$ between these $k$ equality separators can then be formulated as

$$
\begin{aligned}
Vol(\text{decision region of } k \text{ equality separators}) &= Vol(\{\mathbf{x} \in I\}) \\
&= Vol(\cup_{i \in \mathbb{Z}}\{\mathbf{x} \in I : i \le \mathbf{v}^T\mathbf{x} \le i+1\}) \\
&= \sum_{i \in \mathbb{Z}} Vol(\{\mathbf{x} \in I : i \le \mathbf{v}^T\mathbf{x} \le i+1\})
\end{aligned}
$$

because the intersections (hyperplanes) have zero measure. When none of the equality separators are strict equality separators, we obtain an infinite volume since $Vol(\{\mathbf{x} \in I : i \le \mathbf{v}^T\mathbf{x} \le i+1\}) > 0$. By definition of the Lebesgue measure, the intersection has (at least) infinite volume by moving along the $\mathbf{v}$ direction. Otherwise, $Vol(\{\mathbf{x} \in I : i \le \mathbf{v}^T\mathbf{x} \le i+1\}) = 0$, so we have zero volume. Hence, having fewer than $n$ equality separators cannot induce a closed decision region.  $\square$

As an aside, we note that the case where equality separators enclose decision regions is when they are more specifically hyper-hill classifiers (i.e. no label flipping is allowed because normal data belongs within the margin). This is how our insight is birthed that we can simply use hyper-hill classification in AD – we are using a more expressive hypothesis class with equality separation but technically have the VC dimension of hyper-hills (or hyper-ridge classifiers), which is half of equality separators.

### D.2    PROOF OF CLOSING NUMBER OF HALFSPACE SEPARATORS IN $\mathbb{R}^n$

To show that the closing number is at most $n + 1$, we observe that a hypertetrahedron (convex hull of $n + 1$ points or a bounded convex polytope with non-empty interior) can be formed with the intersection of $n + 1$ hyperplanes, where each of the facets of the convex hull correspond to each of the $n + 1$ hyperplanes.

Now, we want to show that $n+1$ is minimal. Since we are considering the intersection of hyperplanes to form a bounded convex polytope with non-empty interior, we assign a one-to-one correspondence between a hyperplane and a facet of the convex polytope. In fact, by considering a dual polytope of the convex polytope, there is a further correspondence to each hyperplane and a vertex of the dual polytope. With non-empty interior, the dual polytope is $n$-dimensional, so has at least $n + 1$ vertices.

Hence, we need at least $n + 1$ facets. Thus, to form a bounded convex polytope with non-empty interior with the intersection of hyperplanes, we need at least $n + 1$ hyperplanes.

## APPENDIX E   BUMP ACTIVATIONS AND LOSS FUNCTIONS

Using different activations and loss functions to model equality separation introduces different implicit biases in the solutions obtained. We introduce possible activations and loss functions, exploring an intuition of the implicit bias induced by these differences (and hence, what applications fit better with these implicit biases).

### E.1   OTHER POSSIBLE BUMP ACTIVATIONS

Other activation functions can be used as a bump in addition to the Gaussian bump activation that we used in this paper. For instance, we can use the inverse-squared of the hyperbolic tangent, $\tanh(v/z^2)$, where $v$ is a hyperparameter controlling the flatness of the bump function near $z = 0$ (Figure 4). Using the plateau can be used as a regularization to drastically reduce the penalization when the input is within some distance from the hyperplane, which is similar to the idea of hinge loss in SVM.

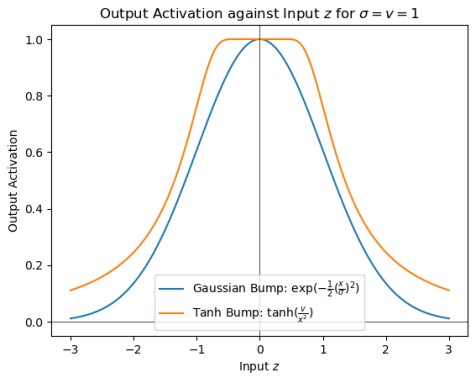
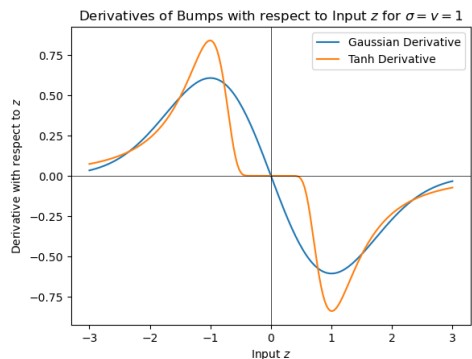

(a) Output of the activation functions.    (b) Derivatives of the activation functions.

Figure 4: Two different bump activations: the Gaussian bump which we used in blue, and the hyperbolic tangent bump in orange.

### E.2   LOSS FUNCTIONS

For input $z$ into the activation function in the output layer, we provide visualizations of how the mean-squared error (MSE) and logistic loss (LL) compare for the negative class (Figure 5a) and positive class (Figure 5b). LL penalizes more on wrong classifications for both classes. Such a scheme is good for learning a stricter decision boundary, but potentially harder to learn when the distributions of the positive and negative class are close and harder when they overlap. On the other hand, MSE has smaller penalization for wrong classifications, which leads to more tolerance for noise but may lead to learning sub-optimal solutions. Such a difference hints at different applications, where LL is generally preferred unless the distributions of the positive and negative class are very close (or overlapping, like in negative sampling), in which case the more conservative MSE would probably be more suitable.

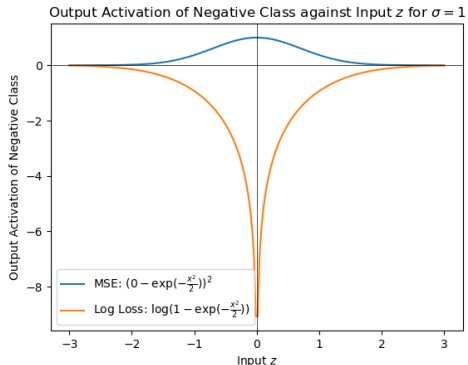 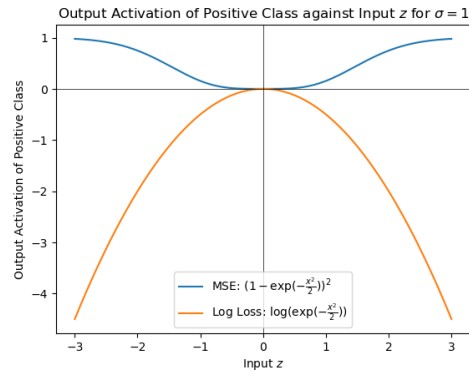

(a) Output of mean-squared error and logistic loss against the input to the bump activation function when the datum is from the negative class.

(b) Output of mean-squared error and logistic loss against the input to the bump activation function when the datum is from the positive class.

Figure 5: Comparison of mean-squared error and logistic loss ($\sigma = 1$ in the bump activation).

## APPENDIX F  EXPERIMENTAL DETAILS

In this section, we provide details of our experiments. Unless otherwise stated, training and testing both have $m = 100$ data samples.

### F.1  ROLE OF $\sigma$ IN THE BUMP ACTIVATION

The gradient of the log-concave bump activation is

$$\frac{\partial}{\partial z} B(z, \mu, \sigma) = -\frac{z - \mu}{\sigma^2} \exp \left[ -\frac{1}{2} \left( \frac{z - \mu}{\sigma} \right)^2 \right], \tag{3}$$

which vanishes to 0 as the input tends towards negative or positive infinity. To prevent vanishing gradients, we initialize $\sigma$ to be decently large to decrease the sensitivity to weight initialization. Experiments in Appendix F.2.3 will shed further light on how the initialization of $\sigma$ affects performance (including the effect of loss functions). For simplicity, we set $\mu = 0$ but give the option for $\sigma$ to be learnable across neurons in the same hidden layer in our experiments. Setting $\sigma$ to be learnable allows the model to adapt to its ability to represent the data, such as decreasing $\sigma^2$ to increase the separation between the two classes. Otherwise, the separation is controlled implicitly by $\mathbf{w}$ through $||\mathbf{w}||_2$, where the margin is proportional to $\frac{\sigma}{||\mathbf{w}||_2}$. Unless otherwise stated, $\sigma$ is assumed to be a fixed hyperparameter at 10. In equality separator NNs, we fix $\sigma = 0.5$ in the output layer for stability during training.

### F.2  LINEAR CLASSIFICATION FOR ANOMALY DETECTION

To understand the usability of equality separation in NNs, we first look at linear classification for AD with (1) linearly inseparable data and (2) linearly separable data. We select key results to show here and include more details in the following subsections.

**Linear problem 1: living on the plane, linearly inseparable**  We generate data on hyperplanes as the positive class while the negative class follows a uniform distribution in the ambient space. The positive to negative class imbalance (P:N) of 0.9:0.1. Weights and bias of the hyperplane are drawn from a standard multivariate normal, and data are perturbed with Gaussian noise centered at zero with varying standard deviations. We repeat experiments 20 times. Results are reported in Table 5.

Especially in higher dimensions, our equality separator is robust even when noise is sampled with higher variance than the weights. The inductive bias of hyperplane learning helps our equality separator to be less sensitive to the signal-to-noise ratio. Although simple linear regression with only positive examples may be more suitable in this case, we demonstrate that using negative examples as regularization can still achieve good performance, suggesting its usability in deep learning.

Table 5: AUPR for linear AD of linearly inseparable data with varying standard deviation (Std) of noise and data dimension (Dim). No noise is denoted by 0.0 std. Random classifier has 0.90 AUPR.

| Std\Dim | 5 | 10 | 15 | 20 | 25 | 30 | 35 |
|---|---|---|---|---|---|---|---|
| 0.0 | 0.99±0.06 | 1.00±0.00 | 0.99±0.06 | 1.00±0.00 | 1.00±0.00 | 0.99±0.06 | 1.00±0.00 |
| 0.5 | 0.98±0.06 | 1.00±0.00 | 0.99±0.06 | 1.00±0.00 | 1.00±0.00 | 0.99±0.06 | 1.00±0.00 |
| 1.0 | 0.98±0.05 | 1.00±0.01 | 0.98±0.05 | 1.00±0.00 | 0.99±0.04 | 0.99±0.06 | 1.00±0.00 |
| 1.5 | 0.97±0.05 | 1.00±0.01 | 0.98±0.05 | 0.99±0.01 | 0.99±0.01 | 0.99±0.06 | 1.00±0.01 |
| 2.0 | 0.97±0.05 | 0.98±0.02 | 0.99±0.01 | 0.99±0.01 | 1.00±0.00 | 0.98±0.05 | 1.00±0.00 |

Table 6: AUPR for 2D Gaussian Separation by SVM and our equality separator (ES), with varying noise multiplier (NM). ES is competitive with SVM.

| NM | 1.0 | 4.0 | 7.0 | 10.0 | 13.0 | 16.0 | 19.0 |
|---|---|---|---|---|---|---|---|
| SVM | 1.00±0.00 | 1.00±0.00 | 0.99±0.01 | 0.98±0.01 | 0.97±0.03 | 0.97±0.01 | 0.97±0.03 |
| ES (ours) | 1.00±0.00 | 1.00±0.00 | 0.99±0.01 | 0.98±0.01 | 0.97±0.02 | 0.97±0.01 | 0.97±0.01 |

**Linear problem 2: competing with halfspace separators on linear separability**  We compare our equality separator with robust halfspace separation, linear-kernel SVM with weighted class loss of the P:N of 0.9:0.1. To vary the overlap of distributions, we multiply the covariance matrix of each Gaussian by varying scalars, denoted as a *noise multiplier*. We repeat experiments 20 times and report results in Table 6.

Unlike the SVM trained, our equality separator does not explicitly account for class imbalance and does not incoporate the robustness of hinge loss. Yet, our equality separator performs as well as the class-weighted SVM. Once again, the bias of hyperplane learning encoded in equality separators makes it robust.

### F.2.1   LINEAR PROBLEM 1: LINEARLY INSEPARABLE DATA

We report the details of the experiments done on our experiments on linearly inseparable data. As mentioned, we model the positive class to live on some hyperplane, while the negative class lives uniformly in the ambient space, where we vary the dimension of the ambient space $d$ across experiments to be $5, 10, 15, 20, 25, 30$ and $35$. The normal vector $\mathbf{w} \in \mathbb{R}^d$ of the hyperplane is drawn from a standard multivariate normal

$$\mathbf{w} \sim \mathcal{MVN}(0, I_d)$$

while the bias $b \in \mathbb{R}$ is drawn from a standard normal

$$b \sim \mathcal{N}(0, 1).$$

A datum from the positive class $\mathbf{x}_p$ is generated where the first $d-1$ dimensions are drawn from a uniform distribution

$$x_p^i \sim \text{Uniform}(-10, 10) \,, i \in \{1, ..., d-1\}$$

and the last $d^{th}$ dimension is calculated by

$$x_p^d = \frac{-b - \sum_{i=1}^{d-1} w_i x_p^i}{w_d}$$

where $w_i \in \mathbb{R}$ is the $i^{th}$ dimension of normal vector $\mathbf{w} \in \mathbb{R}^d$. On the other hand, data from the negative class are obtained by sampling each dimension from a uniform distribution between $-10$ and $10$. There is a 0.9:0.1 positive to negative class imbalance, and a total of 100 data samples.

After all data are generated, random noise is added to the data. The random noise follows a Gaussian distribution with zero mean and standard deviation of $0.0, 0.5, 1.0$ and $1.5$ (we abuse notation and refer to no noise being added as $0.0$ standard deviation) across the different experiments.

We optimize over mean-squared error (MSE) where the hypothesis class is the set of affine predictors $A_d$ and the hyperparameters $\mu = 0$ and $\sigma = 10$ are fixed. The gradient-based optimizer used is Broyden-Fletcher-Goldfarb-Shannon (BFGS), the loss function used in Section F.2 is MSE, and

Table 7: AUPR for linear AD of linearly inseparable data with varying standard deviation (Std) of noise and data dimension (Dim). No noise is denoted by 0.0 std. Random classifier has 0.90 AUPR. (This table includes results from updated experiments done in the main paper.)

| Std\Dim | 5 | 10 | 15 | 20 | 25 | 30 | 35 |
|---|---|---|---|---|---|---|---|
| 0.0 | 0.99±0.06 | 1.00±0.00 | 0.99±0.06 | 1.00±0.00 | 1.00±0.00 | 0.99±0.06 | 1.00±0.00 |
| 0.5 | 0.98±0.06 | 1.00±0.00 | 0.99±0.06 | 1.00±0.00 | 1.00±0.00 | 0.99±0.06 | 1.00±0.00 |
| 1.0 | 0.98±0.05 | 1.00±0.01 | 0.98±0.05 | 1.00±0.00 | 0.99±0.04 | 0.99±0.06 | 1.00±0.00 |
| 1.5 | 0.97±0.05 | 1.00±0.01 | 0.98±0.05 | 0.99±0.01 | 0.99±0.01 | 0.99±0.06 | 1.00±0.01 |
| 2.0 | 0.97±0.05 | 0.98±0.02 | 0.99±0.01 | 0.99±0.01 | 1.00±0.00 | 0.98±0.05 | 1.00±0.00 |

Table 8: AUPR for linear AD of linearly inseparable data with varying standard deviation (Std) of noise and data dimension (Dim), optimized with mean-squared error. No noise is denoted by 0.0 std. Random classifier has 0.90 AUPR. Number of data samples is 20 times the number of dimensions.

| Std\Dim | 5 | 10 | 15 | 20 | 25 | 30 | 35 |
|---|---|---|---|---|---|---|---|
| 0.0 | 0.99±0.06 | 1.00±0.00 | 1.00±0.00 | 1.00±0.00 | 1.00±0.00 | 0.99±0.05 | 1.00±0.00 |
| 0.5 | 0.98±0.06 | 1.00±0.00 | 0.98±0.05 | 1.00±0.00 | 1.00±0.00 | 0.98±0.05 | 1.00±0.00 |
| 1.0 | 0.98±0.05 | 0.99±0.01 | 0.99±0.00 | 0.99±0.01 | 0.99±0.00 | 0.98±0.05 | 0.99±0.01 |
| 1.5 | 0.97±0.05 | 0.99±0.00 | 0.98±0.05 | 0.99±0.01 | 0.99±0.01 | 0.97±0.05 | 0.99±0.01 |
| 2.0 | 0.97±0.05 | 0.99±0.01 | 0.98±0.01 | 0.98±0.01 | 0.98±0.01 | 0.97±0.05 | 0.98±0.01 |

the parameters are initialized to a standard multivariate normal distribution. We use the $\mathrm{minimize}$ function in the $\mathrm{scipy.optimize}$ package where the stopping criteria is when the step size is $0$.[9]

Each trial is repeated a total of 20 times by generating a new dataset by sampling a different weight and bias vector (that defines a different hyperplane) and running optimization again, and we report the mean and standard deviation of the results. Updated results are reported in Table 7.

The results of our equality separator reported are somewhat interesting. On one hand, as expected, an increased level of noise leads to a decreased mean AUPR. With more noise, the overlap between the positive and negative class increases, so the separation among the classes (and hence the mean AUPR) decreases. On the other hand, the performance in higher dimensions generally increases, with mean AUPR increasing and AUPR variance decreasing. This phenomenon is probably because there is a greater density of noise in lower dimensions for constant amount of data, leading to a greater overlap of the 2 classes.

Furthermore, a linear increase in the data does not seem to affect the performance significantly too. We performed the same experiment, but instead of a constant 100 data samples for all experiments, the number of data samples was adjusted be $20d$ for input dimension $d$. Increasing the number of data samples will increase the amount of data from the negative class, and hence increase the overlap between the positive and negative class. However, results reported in Table 8 suggest that this increase in overlap does not affect the mean and variance of AUPR of the equality separator. The inductive bias of linear regression helps in hyperplane learning, even when the overlap between the positive and negative class increases.

As seen in both Table 7 and Table 8, there are instances where results are reported with high variance (particularly in 5D and 30D data). These instances of high variability of results suggest that the equality separator could be sensitive to initialization. Such variability would be decreased in a regular least-squares linear regression because gradient descent can be run on the convex mean-squared error loss and without the need for including regularization of the negative examples. However, the regularization in the equality separator to have negative examples further away from the hyperplane means that the equality separator is able to use negative examples and is useful in deep methods for supervised anomaly detection.

---

[9]https://docs.scipy.org/doc/scipy/reference/optimize.minimize-bfgs.html#optimize-minimize-bfgs

Table 9: Logistic loss optimization with constant amount of data for linear AD of linearly inseparable data with varying standard deviation (Std) of noise and data dimension (Dim). No noise is denoted by 0.0 std. Constant 100 data samples. AUPR is reported.

| Dim\Std | 5 | 10 | 15 | 20 | 25 | 30 | 35 |
|---|---|---|---|---|---|---|---|
| 0.0 | 0.95±0.06 | 0.98±0.01 | 0.98±0.01 | 0.99±0.0 | 0.99±0.01 | 0.98±0.05 | 1.00±0.00 |
| 0.5 | 0.96±0.05 | 0.98±0.01 | 0.98±0.01 | 0.99±0.00 | 0.99±0.01 | 0.98±0.05 | 1.00±0.00 |
| 1.0 | 0.95±0.05 | 0.98±0.02 | 0.98±0.01 | 0.99±0.0 | 0.99±0.01 | 0.98±0.05 | 1.00±0.00 |
| 1.5 | 0.94±0.05 | 0.97±0.02 | 0.97±0.01 | 0.98±0.01 | 0.99±0.01 | 0.98±0.05 | 1.00±0.00 |
| 2.0 | 0.93±0.05 | 0.96±0.03 | 0.97±0.01 | 0.98±0.01 | 0.98±0.01 | 0.97±0.05 | 0.99±0.01 |

Table 10: Logistic loss optimization with number of samples 20 times the number of dimensions, done for linear AD of linearly inseparable data with varying standard deviation (Std) of noise and data dimension (Dim). No noise is denoted by 0.0 std. Number of data samples is 20 times the number of dimensions. AUPR is reported.

| Std\Dim | 5 | 10 | 15 | 20 | 25 | 30 | 35 |
|---|---|---|---|---|---|---|---|
| 0.0 | 0.95±0.06 | 0.96±0.01 | 0.96±0.01 | 0.96±0.01 | 0.96±0.01 | 0.96±0.01 | 0.96±0.01 |
| 0.5 | 0.96±0.05 | 0.96±0.02 | 0.95±0.01 | 0.96±0.01 | 0.96±0.01 | 0.95±0.01 | 0.96±0.01 |
| 1.0 | 0.95±0.05 | 0.96±0.02 | 0.95±0.01 | 0.96±0.02 | 0.96±0.01 | 0.96±0.01 | 0.95±0.01 |
| 1.5 | 0.94±0.05 | 0.95±0.02 | 0.95±0.02 | 0.95±0.01 | 0.95±0.01 | 0.95±0.01 | 0.95±0.02 |
| 2.0 | 0.93±0.05 | 0.95±0.02 | 0.94±0.02 | 0.95±0.01 | 0.95±0.02 | 0.94±0.02 | 0.94±0.01 |

**Logistic loss** To compare our results with MSE, we also use LL and report results in Table 9 for constant 100 data samples across experiments and Table 10 for the same linear increase in amount of data as explored in the MSE case. For constant data, we also observe that LL produces performance that improves (higher mean and smaller variance in AUPR) as the number of dimensions increases. On the other hand, the same linear increase in data as the MSE case does not increase the mean AUPR for equality separators optimized with LL, but only decreases the variance of the AUPR. This simple comparison hints at the different usages of different loss functions.

As suggested in Appendix E.2, LL is more sensitive to wrong classifications, so it performs worse as the overlap between the positive and negative class increases (ie. as the noise increases). Since noise is sparser in higher dimension for constant data settings, there is less overlap and LL performs better as the dimensions increase. Since the hyperplane is overdetermined with the data, finding the hyperplane is not much of a problem. However, when the amount of data increases, the amount of noise also increases. Since LL is more sensitive than MSE to the overlap between the positive and negative classes, the higher density of noise causes LL to degrade in performance.

### F.2.2 LINEARLY SEPARABLE DATA

We report the details of the experiments done for linearly separable data. As mentioned, the each of the two classes are sampled from a 2D multivariate Gaussians. The positive class follows

$$\mathbf{x}_{positive} \sim \mathcal{MVN}\left(\begin{bmatrix} 1 \\ 1 \end{bmatrix}, k \begin{bmatrix} 0.2 & 0.1 \\ 0.1 & 0.2 \end{bmatrix}\right)$$

while the negative class follows

$$\mathbf{x}_{negative} \sim \mathcal{MVN}\left(\begin{bmatrix} -1 \\ -1 \end{bmatrix}, k \begin{bmatrix} 0.2 & 0.1 \\ 0.1 & 0.2 \end{bmatrix}\right)$$

where $k$ is the *noise multiplier*, a scalar that determines the spread of each class (and hence, the overlap between both classes). There is a 0.9:0.1 P:N imbalance and a total of 100 data samples for both training and testing datasets.

**SVM**  To implement the SVM, we use the sklearn.SVM.SVC support vector classifier class in the scikit-learn package. We specify a linear kernel and balanced class weight in its argument, and the prediction probability is based on an internal 5-fold cross validation.[10]

**Equality Separator**  We optimize over mean-squared error as in Equation 2 where the hypothesis class is the set of affine predictors $A_2$ and the hyperparameters $\mu = 0$ and $\sigma = 10$ are fixed. The gradient-based optimizer used is BFGS, and the parameters are initialized to a standard multivariate normal distribution. We use the minimize function in the scipy.optimize package where the stopping criteria is when the step size is $0$.

Each trial is repeated a total of 20 times by generating a new dataset by re-sampling from the same multivariate Gaussians and running optimization again, and we report the mean and standard deviation of the results.

### F.2.3  STUDY ON SENSITIVITY OF $\sigma$

We repeat experiments done in Section F.2 on linear separable data, but vary the value of $\sigma$ in the bump activation (Equation 2) to $0.1, 0.5, 5$ and $10$. We report results in Table 11.

Equality separators with $\sigma \geq 0.5$ have competitive performance with SVMs, and results are mostly within 1 standard deviation away from each other across the equality separators with $\sigma$ values of $0.5, 5$ and $10$ (Figure 6a). Equality separators with a small $\sigma$ at $0.1$ have slightly worse performance in the low noise settings, with a lower mean and higher variance for its AUPR, but quickly become competitive with SVMs a noise multiplier of 2.0 onwards. These good results for large $\sigma$ suggest that large $\sigma$ values decrease the sensitivity to randomness, such as the initialization of the weights in the equality separator. These empirical results corroborate with our comment in Appendix F.1 about vanishing gradients with small $\sigma$ values impeding effective gradient-based optimization. However, larger $\sigma$ values widen the bump of the Gaussian, which increases the margin for error and may cause predictions to be less informative.

Additionally, for all values of $\sigma$, equality separation is still quite robust at high noise levels, obtaining a slightly lower AUPR variance while maintaining the same mean AUPR as SVM. Lower variance in AUPR suggests that equality separators can potentially be less sensitive to noise than SVMs. The decreased sensitivity could be due to the equality separator modelling the hyperplane of the normal class (positive class) rather than the separating hyperplane, capitalizing on the class with more data. Hence, equality separation in class-imbalanced settings can produce competitive results with SVMs, especially when the overlap between the positive and negative class increases.

**Logistic loss**  We perform the same experiments on linearly separable data, but change the loss function to LL. We report results in Table 12 and provide a visualization in Figure 6b.

Training on LL is generally slightly less competitive than training with MSE, although not significantly less. With an increase in overlap between the positive and negative classes, there is generally a slightly faster decrease in mean AUPR, with a higher AUPR variance. As explained in Appendix E.2, LL does not handle overlap between classes well, leading to a significant decrease in performance. Interestingly enough, smaller $\sigma$ values seem to achieve better performance with LL. Smaller $\sigma$ values produce stricter decision boundaries, similar to LL compared to MSE, which suggest how loss functions and $\sigma$ values may pair well together.

**Varying dimensionality of linearly separable data**  To understand the effect of increasing dimensionality, we also vary the dimensionality $d$ of the multivariate Gaussian. For simplicity, we generate data from zero mean isotropic Gaussians with covariance $0.1k \cdot I_d$ for noise multiplier $k$. We test $d = 5, 10, 15, 20, 25, 30$. All other details remain the same. As informed by our experiments on varying $\sigma$ for different loss functions, we test on MSE optimization with $\sigma = 10$ and LL optimization with $\sigma = 0.5$.

In general, equality separators are not significantly worse than SVMs, maintaining close to perfect AUPR above 0.99 mean AUPR. As we have observed previously, MSE optimization produces better results (higher mean AUPR with lower variance) in lower dimensions while LL optimization produces

---

[10]https://scikit-learn.org/stable/modules/generated/sklearn.svm.SVC.html

Table 11: AUPR for 2D Gaussian separation by SVM and our equality separator (ES), with varying noise multiplier (NM) optimized with mean-squared error. The value of $\sigma$ in the bump activation is varied in the ES to illustrate its effect.

| NM | SVM | ES ($\sigma = 0.1$) | ES ($\sigma = 0.5$) | ES ($\sigma = 5.0$) | ES ($\sigma = 10.0$) |
|---|---|---|---|---|---|
| 1.0 | 1.00±0.0 | 0.96±0.08 | 1.00±0.00 | 1.00±0.00 | 1.00±0.00 |
| 1.5 | 1.00±0.0 | 0.99±0.04 | 1.00±0.00 | 1.00±0.00 | 1.00±0.00 |
| 2.0 | 1.00±0.0 | 1.00±0.00 | 1.00±0.00 | 1.00±0.00 | 1.00±0.00 |
| 2.5 | 1.00±0.0 | 1.00±0.00 | 1.00±0.00 | 1.00±0.00 | 1.00±0.00 |
| 3.0 | 1.00±0.0 | 1.00±0.00 | 1.00±0.00 | 1.00±0.00 | 1.00±0.00 |
| 3.5 | 1.00±0.0 | 1.00±0.01 | 1.00±0.01 | 1.00±0.01 | 1.00±0.01 |
| 4.0 | 1.00±0.0 | 1.00±0.00 | 1.00±0.00 | 1.00±0.00 | 1.00±0.00 |
| 4.5 | 0.99±0.0 | 0.99±0.00 | 0.99±0.00 | 0.99±0.00 | 0.99±0.00 |
| 5.0 | 0.99±0.0 | 0.99±0.00 | 0.99±0.00 | 0.99±0.00 | 0.99±0.00 |
| 5.5 | 0.99±0.0 | 0.99±0.01 | 0.99±0.01 | 0.99±0.01 | 0.99±0.01 |
| 6.0 | 0.99±0.01 | 0.99±0.01 | 0.99±0.01 | 0.99±0.01 | 0.99±0.01 |
| 6.5 | 0.99±0.01 | 0.99±0.01 | 0.99±0.01 | 0.99±0.01 | 0.99±0.01 |
| 7.0 | 0.99±0.01 | 0.99±0.01 | 0.99±0.01 | 0.99±0.01 | 0.99±0.01 |
| 7.5 | 0.99±0.01 | 0.98±0.01 | 0.98±0.01 | 0.98±0.01 | 0.98±0.01 |
| 8.0 | 0.99±0.01 | 0.98±0.01 | 0.98±0.01 | 0.98±0.01 | 0.98±0.01 |
| 8.5 | 0.99±0.01 | 0.98±0.02 | 0.98±0.02 | 0.98±0.02 | 0.98±0.02 |
| 9.0 | 0.99±0.01 | 0.98±0.02 | 0.98±0.02 | 0.98±0.02 | 0.98±0.02 |
| 9.5 | 0.98±0.01 | 0.98±0.01 | 0.98±0.01 | 0.98±0.01 | 0.98±0.01 |
| 10.0 | 0.98±0.01 | 0.98±0.01 | 0.98±0.01 | 0.98±0.01 | 0.98±0.01 |
| 10.5 | 0.98±0.01 | 0.98±0.01 | 0.98±0.01 | 0.98±0.01 | 0.98±0.01 |
| 11.0 | 0.98±0.01 | 0.98±0.01 | 0.98±0.01 | 0.98±0.01 | 0.98±0.01 |
| 11.5 | 0.98±0.01 | 0.98±0.01 | 0.98±0.01 | 0.98±0.01 | 0.98±0.01 |
| 12.0 | 0.98±0.01 | 0.98±0.02 | 0.98±0.02 | 0.98±0.02 | 0.98±0.02 |
| 12.5 | 0.98±0.01 | 0.97±0.02 | 0.97±0.02 | 0.97±0.02 | 0.97±0.02 |
| 13.0 | 0.97±0.03 | 0.97±0.02 | 0.97±0.02 | 0.97±0.02 | 0.97±0.02 |
| 13.5 | 0.97±0.03 | 0.97±0.02 | 0.97±0.02 | 0.97±0.02 | 0.97±0.02 |
| 14.0 | 0.97±0.03 | 0.97±0.02 | 0.97±0.02 | 0.97±0.02 | 0.97±0.02 |
| 14.5 | 0.98±0.01 | 0.97±0.01 | 0.97±0.01 | 0.97±0.01 | 0.97±0.01 |
| 15.0 | 0.98±0.01 | 0.97±0.01 | 0.97±0.01 | 0.97±0.01 | 0.97±0.01 |
| 15.5 | 0.97±0.03 | 0.97±0.01 | 0.97±0.01 | 0.97±0.01 | 0.97±0.01 |
| 16.0 | 0.97±0.01 | 0.97±0.01 | 0.97±0.01 | 0.97±0.01 | 0.97±0.01 |
| 16.5 | 0.97±0.01 | 0.97±0.02 | 0.97±0.02 | 0.97±0.02 | 0.97±0.02 |
| 17.0 | 0.97±0.03 | 0.97±0.01 | 0.97±0.01 | 0.97±0.01 | 0.97±0.01 |
| 17.5 | 0.97±0.03 | 0.97±0.01 | 0.97±0.01 | 0.97±0.01 | 0.97±0.01 |
| 18.0 | 0.97±0.03 | 0.97±0.01 | 0.97±0.01 | 0.97±0.01 | 0.97±0.01 |
| 18.5 | 0.97±0.03 | 0.97±0.01 | 0.97±0.01 | 0.97±0.01 | 0.97±0.01 |
| 19.0 | 0.97±0.03 | 0.97±0.01 | 0.97±0.01 | 0.97±0.01 | 0.97±0.01 |
| 19.5 | 0.97±0.03 | 0.97±0.01 | 0.97±0.01 | 0.97±0.01 | 0.97±0.01 |
| 20.0 | 0.97±0.03 | 0.97±0.01 | 0.97±0.01 | 0.97±0.01 | 0.97±0.01 |

Table 12: AUPR for 2D Gaussian separation by SVM and our equality separator (ES), with varying noise multiplier (NM) optimized with logistic loss. The value of $\sigma$ in the bump activation is varied in the ES to illustrate its effect.

| NM | SVM | ES ($\sigma = 0.1$) | ES ($\sigma = 0.5$) | ES ($\sigma = 5.0$) | ES ($\sigma = 10.0$) |
|---|---|---|---|---|---|
| 1.0 | 1.00±0.00 | 0.99±0.04 | 1.00±0.00 | 1.00±0.00 | 1.00±0.00 |
| 1.5 | 1.00±0.00 | 0.98±0.04 | 1.00±0.00 | 1.00±0.00 | 0.99±0.03 |
| 2.0 | 1.00±0.00 | 1.00±0.01 | 1.00±0.00 | 1.00±0.00 | 0.99±0.02 |
| 2.5 | 1.00±0.00 | 0.99±0.01 | 1.00±0.00 | 0.99±0.02 | 1.00±0.01 |
| 3.0 | 1.00±0.00 | 0.99±0.01 | 1.00±0.00 | 0.99±0.02 | 0.99±0.02 |
| 3.5 | 1.00±0.00 | 0.99±0.03 | 1.00±0.00 | 0.98±0.02 | 0.98±0.02 |
| 4.0 | 1.00±0.00 | 0.98±0.02 | 0.99±0.02 | 0.98±0.02 | 0.97±0.03 |
| 4.5 | 0.99±0.00 | 0.99±0.01 | 0.98±0.02 | 0.98±0.02 | 0.97±0.04 |
| 5.0 | 0.99±0.00 | 0.99±0.01 | 0.97±0.03 | 0.97±0.03 | 0.97±0.03 |
| 5.5 | 0.99±0.00 | 0.98±0.02 | 0.97±0.03 | 0.97±0.03 | 0.97±0.03 |
| 6.0 | 0.99±0.01 | 0.98±0.03 | 0.99±0.01 | 0.96±0.03 | 0.96±0.03 |
| 6.5 | 0.99±0.01 | 0.99±0.01 | 0.97±0.03 | 0.96±0.03 | 0.95±0.04 |
| 7.0 | 0.99±0.01 | 0.98±0.02 | 0.96±0.05 | 0.97±0.03 | 0.95±0.03 |
| 7.5 | 0.99±0.01 | 0.98±0.02 | 0.98±0.02 | 0.97±0.02 | 0.95±0.03 |
| 8.0 | 0.99±0.01 | 0.99±0.01 | 0.97±0.03 | 0.97±0.02 | 0.95±0.03 |
| 8.5 | 0.99±0.01 | 0.97±0.03 | 0.97±0.02 | 0.96±0.03 | 0.94±0.04 |
| 9.0 | 0.99±0.01 | 0.98±0.02 | 0.98±0.01 | 0.97±0.03 | 0.95±0.04 |
| 9.5 | 0.98±0.01 | 0.98±0.01 | 0.97±0.03 | 0.96±0.04 | 0.95±0.03 |
| 10.0 | 0.98±0.01 | 0.97±0.03 | 0.98±0.01 | 0.96±0.03 | 0.95±0.03 |
| 10.5 | 0.98±0.01 | 0.96±0.03 | 0.97±0.03 | 0.95±0.04 | 0.95±0.03 |
| 11.0 | 0.98±0.01 | 0.97±0.02 | 0.97±0.02 | 0.95±0.03 | 0.95±0.03 |
| 11.5 | 0.98±0.01 | 0.97±0.02 | 0.97±0.03 | 0.95±0.04 | 0.95±0.03 |
| 12.0 | 0.98±0.01 | 0.98±0.02 | 0.98±0.01 | 0.95±0.03 | 0.94±0.03 |
| 12.5 | 0.98±0.01 | 0.97±0.02 | 0.96±0.02 | 0.95±0.03 | 0.94±0.03 |
| 13.0 | 0.97±0.03 | 0.96±0.03 | 0.95±0.05 | 0.95±0.03 | 0.95±0.03 |
| 13.5 | 0.97±0.03 | 0.97±0.02 | 0.96±0.03 | 0.95±0.03 | 0.95±0.03 |
| 14.0 | 0.97±0.03 | 0.97±0.02 | 0.96±0.03 | 0.95±0.03 | 0.95±0.03 |
| 14.5 | 0.98±0.01 | 0.97±0.02 | 0.97±0.02 | 0.95±0.03 | 0.95±0.03 |
| 15.0 | 0.98±0.01 | 0.97±0.02 | 0.97±0.02 | 0.95±0.03 | 0.95±0.03 |
| 15.5 | 0.97±0.03 | 0.97±0.02 | 0.97±0.02 | 0.95±0.03 | 0.94±0.03 |
| 16.0 | 0.97±0.01 | 0.97±0.02 | 0.96±0.03 | 0.94±0.04 | 0.94±0.03 |
| 16.5 | 0.97±0.01 | 0.96±0.03 | 0.95±0.03 | 0.94±0.03 | 0.95±0.03 |
| 17.0 | 0.97±0.03 | 0.96±0.03 | 0.95±0.03 | 0.95±0.03 | 0.95±0.03 |
| 17.5 | 0.97±0.03 | 0.96±0.03 | 0.97±0.02 | 0.95±0.03 | 0.95±0.03 |
| 18.0 | 0.97±0.03 | 0.96±0.03 | 0.95±0.04 | 0.95±0.03 | 0.94±0.03 |
| 18.5 | 0.97±0.03 | 0.96±0.02 | 0.97±0.02 | 0.95±0.03 | 0.94±0.03 |
| 19.0 | 0.97±0.03 | 0.96±0.03 | 0.96±0.03 | 0.95±0.03 | 0.95±0.03 |
| 19.5 | 0.97±0.03 | 0.96±0.02 | 0.95±0.04 | 0.95±0.03 | 0.94±0.03 |
| 20.0 | 0.97±0.03 | 0.95±0.04 | 0.95±0.04 | 0.95±0.03 | 0.94±0.03 |

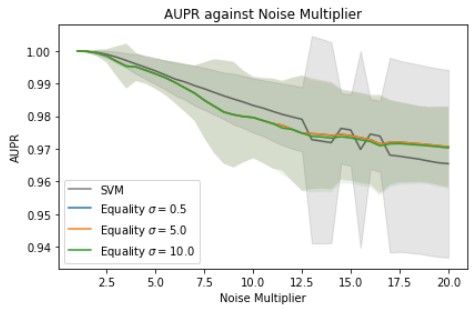 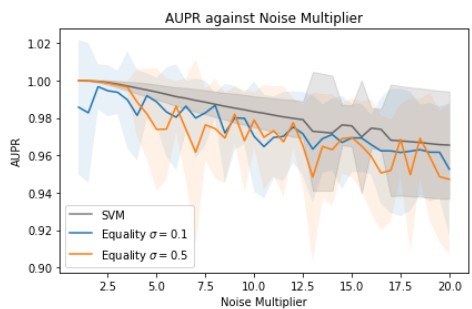

(a) Mean-squared error optimization. $\sigma = 0.1$ omitted for clearer visualizations.

(b) Logistic loss optimization. $\sigma = 5.0, 10.0$ omitted for clearer visualizations.

Figure 6: AUPR for 2D Gaussian separation by SVM and equality separator under increasing noise and $\sigma$ in its bump activation. Mean-squared error and logistic loss optimization are shown.

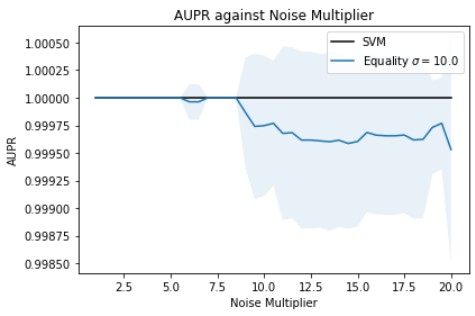 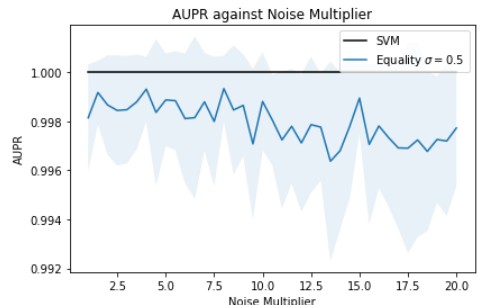

(a) Mean-squared error optimization with $\sigma = 10$.

(b) Logistic loss optimization with $\sigma = 0.5$.

Figure 7: AUPR of equality separators and SVMs on two linearly separable 30D multivariate Gaussians.

better results in higher dimensions. Figure 7 is a sample visualization of the comparison between the performance of equality separators and SVMs at $d = 30$.

### F.3 NON-LINEAR ANOMALY DETECTION: SYNTHETIC DATA

We report additional details of the experiments done in Section 4.1. Experiments are repeated 20 times by re-initializing models and training them on the dataset. By using the same dataset but re-initializing the models, we can observe each model's sensitivity to initializations and optimization methods. We perform experiments with a relatively low positive to negative class imbalance of 0.75:0.25 during train and test time to more prominently compare the difference between models.

As a note, we refer to the normal class as the positive class in the text because it corresponds to a neuronal activation. However, when calculating results, we follow the literature and refer to anomalies as the positive class. Since we focus on comparing between the *separation* between normal data and anomalies across methods, these details do not affect the analysis.

**Shallow models** Shallow models are trained using the sklearn library and the boosted trees from the XGBoost library[11]. Hyperparameters are picked using 10-fold stratified cross validation under logistic loss on the validation dataset. We list the hyperparameters that were chosen. If a hyperpameter is not listed, we used the default ones from the respective libraries.

- Decision trees: Split with random split and half of the features, maximum depth of the tree at 8, no maximum leaf nodes, minimum number of samples required to split an internal node and to be at a leaf node at 2 and 1 respectively, and no class weight.

---
[11]https://xgboost.readthedocs.io/en/stable/python/python_api.html

Table 13: Test AUPR for equality separator with RBF kernel for non-linear anomaly detection. We vary the loss function (mean-squared error or logistic loss) and the value of $\sigma$ $(0.1, 0.5, 5, 10)$

| Loss\$\sigma$ | 0.1 | 0.5 | 5 | 10 |
|---|---|---|---|---|
| MSE | 0.98±0.04 | 1.00±0.01 | **1.00±0.00** | **1.00±0.00** |
| LL | 0.99±0.03 | **1.00±0.00** | **1.00±0.00** | 1.00±0.01 |

- Random forests: Split with half of the features, 50 trees, no maximum depth of a tree, no maximum leaf nodes, minimum number of samples required to split an internal node and to be at a leaf node at 2 and 1 respectively, and no class weight.

- XGBoost (extreme gradient boosted trees): 50 trees, 0 logistic loss reduction required for further splits, $\alpha = 0.1$ for $\ell_1$ regularization, $\lambda = 0.01$ for $\ell_2$ regularization, balanced class weight and subsampling ratio of 0.6.

- Logistic regression with RBF kernel: $C = 1/\lambda = 0.01$ regularization parameter with $\ell_2$ penalty and no class weight.

- SVM with RBF kernel: $C = 1/\lambda = 0.1$ regularization parameter with $\ell_2$ penalty and no class weight.

- OCSVM with RBF kernel: The upper bound set on the fraction of training errors is 0.0001, the kernel coefficient is the inverse of the product of the number of features (2) and the variance of the data.

- Isolation forest: 50 trees that use all samples and all features and an automatic detection of the contamination ratio based in the original paper (Liu et al., 2008).

- Local outlier factor: 5 neighbours used with a leaf size of 30 and an automatic detection of the contamination ratio based in the original paper (Breunig et al., 2000).

For unsupervised AD, there are two possible ways to use the labels: we can either feed all the data into the algorithm (and include information of the ratio of anomalies to normal data when possible, such as for OCSVM), or we can just feed in normal data into the algorithm and remove the anomalies (because unsupervised methods assume that most/all data fed in is normal). We noticed that the first method gives us better results, and we report these.

For equality separators, we report the results of varying the loss function (LL or MSE) and the value of $\sigma$ $(0.1, 0.5, 5, 10)$ in Table 13. Equality separators optimized with LL performed better with lower values of $\sigma$, while those optimized with MSE performed better with greater values of $\sigma$. When $\sigma = 0.1$, the equality separators had a tendency of underfitting, scoring an lower average AUPR on train data than that of test data. The underfitting phenomenon at this low $\sigma$ value corroborates with our observation of the sensitivity of initialization with smaller $\sigma$ values to prevent vanishing gradients.

**Neural networks** NNs used are fully connected feed-forward NNs. Below are the details of the NNs:

1. NNs are trained using TensorFlow.[12] All parameters are assumed to be default unless stated below.

2. We use 5 neurons per hidden layer, limiting the chance of overfitting in this low data regime.

3. NNs with leaky ReLU activations in their hidden layers have the leaky ReLU with parameter 0.01. Leaky ReLU is chosen over the conventional ReLU because mappings that are more local (such as the bump activation) are more sensitive to initializations (Flake, 1993), and backpropagation with leaky ReLU will not suffer from dead neurons like ReLU would. Since we are creating relatively small NNs, the impact of dead neurons may potentially be large. The parameter is chosen to be relatively small at 0.01 to simulate ReLU more closely.

4. NNs that have bump activations in their hidden layers are initialized with $\sigma = 0.5$, which is not trainable by default.

---

[12]https://www.tensorflow.org/

5. NNs with RBF activations in their hidden layers or output layer have a fixed weight of 1.0 with learnable centers that are initialized with Glorot uniform initializer. Fixed weights with learnable centers make a fair comparison so that NNs will have the same number of parameters.[13]

6. Equality separators have an output of a bump activation with $\sigma = 0.5$ that is fixed at $\sigma = 0.5$ by default and have a default loss function of binary cross entropy (ie. logistic loss in this binary classification problem) that can also be switched out for mean-squared error, while halfspace separators have an sigmoid output activation and have a loss function of binary cross entropy.

7. Weights are initialized with the default Glorot uniform initializer.

8. NNs are trained with the Adam optimizer under a constant learning rate of $0.001$.

9. Training is done for 1000 epochs with early stopping. Early stopping monitors validation loss with a patience of 10 and best weights are restored. Validation split is 0.1.

In addition to the results presented in the main paper, we include results for equality separators optimized with MSE and equality separators with bump activations in their hidden layers with learnable $\sigma$. AUPRs of the test dataset are reported in Table 14. From our results, equality separators with RBF activations perform the best with a perfect AUPR during train and test times, regardless whether they were optimized under LL or MSE.

Furthermore, as previously discussed, we observe a larger separation between the positive class and the negative test class for these equality separators with RBF activations optimized under LL. Looking at individual models, about half of these equality separators achieved high separation between the normal and anomaly class, where the difference between the mean output of normal data and anomalies during testing is at least 0.30 (Figure 3f). In the penultimate layer, this separation corresponds to normal data being a distance within about 0.05 away from the hyperplane, while anomalies are a distance of about 0.42 to 0.60 away. Even though the output of anomalies is still relatively high (above 0.5), from the equality separation perspective, these anomalies are more than 8 times further than normal data, so setting reliable thresholds is more straightforward.

The difference between the means of the positive class and negative class during test time was $0.27 \pm 0.11$ for equality separators with RBF activations optimized under LL, compared to $0.16 \pm 0.11$ for those optimized under MSE. In the penultimate layer of the NNs, the average difference corresponds to a distance of the negative class from the hyperplane of 0.40 for LL optimization and 0.30 for MSE optimization (an output prediction of 0.5 corresponds to a distance of 0.59 away from the hyperplane). Since this experiment had no overlap between the positive and negative class, LL optimization was more appropriate, obtaining perfect AUPR like MSE optimization but a greater separation between the positive and negative class. A greater separation suggests one-class decision boundaries that are better.

The next best models achieving close to perfect AUPR are the equality separators with bump activations and RBF separators with leaky ReLU or RBF activations. NNs with bump activations in their hidden layers consistently achieved mean AUPRs of above 0.90, suggesting the versality of the bump activation as a semi-local activation function.

On a whole, it seems that a mix of globality and locality in activation functions helps to increase one-class classification performance. While equality separation with the local RBF activation was the most effective, equality separators with bump activations (purely semi-local) and RBF separators with leaky ReLU (local paired with global) also have competitive AUPR. When using leaky ReLU in the hidden layers, increasing the locality of the output activation increases performance, while the less global bump and RBF activations also increase performance of halfspace separators. Meanwhile, NNs with bump activations in their hidden layers consistently achieved over 0.90 mean AUPR. Our experiments corroborate with our intuition in Section 3.3 that more local mappings (such as the equality separator and its smooth version, the bump activation) can implicitly encode a one-class

---

[13] Note that each neuron from a regular dense layer will have a total $d + 1$ parameters in the affine scaling $\mathbf{w}^T \mathbf{z} + b$ for $\mathbf{w}, \mathbf{z} \in \mathbb{R}^d$, but the distance from the hyperplane is determined by the degrees of freedom, which will still be $d$. Meanwhile, the center parameter in an RBF neuron is in the same space as the input $\mathbf{z}$, so it is $d$-dimensional. Hence, we view the number of parameters in a single RBF layer and a single fully connected layer to be the same.

Table 14: Test time AUPR for supervised AD using neural networks with 2 hidden layers. For NNs, we modify the output layer (halfspace separation (HS), equality separation (ES) and RBF separation (RS)) in the rows and activation functions in the columns. For equality separators, we include information on the loss optimized on (logistic loss LL or mean-squared error MSE). If $\sigma$ is learnable in the bump activations in the hidden layers, we include an added "s" suffix. (This table includes results from updated experiments done in the main paper.)

| NN\Activation | Leaky ReLU | Bump | RBF |
| --- | --- | --- | --- |
| HS | 0.62±0.12 | 0.92±0.13 | 0.83±0.18 |
| RS | 0.97±0.05 | 0.91±0.14 | 0.97±0.06 |
| ES (LL) | 0.83±0.15 | 0.99±0.04 | **1.00±0.00** |
| ES (MSE) | 0.77±0.18 | 0.98±0.09 | **1.00±0.00** |
| ES-s (LL) | - | 0.98±0.05 | - |
| ES-s (MSE) | - | 0.95±0.11 | - |

Table 15: Test time AUPR for supervised AD using neural networks with 3 hidden layers. For NNs, we modify the output layer (halfspace separation (HS), equality separation (ES) and RBF separation (RS)) in the rows and activation functions in the columns. For equality separators, we include information on the loss optimized on (logistic loss LL or mean-squared error MSE). If $\sigma$ is learnable in the bump activations in the hidden layers, we include an added "s" suffix.

| NN\Activation | Leaky ReLU | Bump | RBF |
| --- | --- | --- | --- |
| HS | 0.69±0.16 | 0.92±0.14 | 0.63±0.17 |
| RS | 0.96±0.06 | 0.89±0.14 | 0.94±0.13 |
| ES (LL) | 0.83±0.15 | 0.95±0.10 | **1.00±0.00** |
| ES (MSE) | 0.84±0.16 | 0.90±0.13 | 0.99±0.03 |
| ES-s (LL) | - | 0.98±0.04 | - |
| ES-s (MSE) | - | 0.90±0.16 | - |

classification objective. The semi-local property of equality separators allows it to retain locality for AD, while improving the separation between the normal and anomaly class of NNs with only RBF activations.

**3 hidden layers**   We also investigate NNs with 3 hidden layers, which is 1 more hidden layer than our original experiments. We report test time AUPRs in Table 15. In general, NNs with 3 hidden layers achieved worse performance than their counterparts with 2 hidden layers. Coupled with train AUPRs of above 0.97, we can see that NNs with 3 hidden layers generally overfit to the negative class during training. Equality separator NNs with RBF activations with 3 hidden layers optimized under LL maintained a perfect AUPR and achieved a separation between the positive and negative class during test time of $0.22 \pm 0.13$, which has a mean that is only slightly lower than their counterpart with 2 hidden layers. Those optimized under MSE also maintained a close to perfect AUPR with a separation of $0.16 \pm 0.14$, which is similar to their counterparts with 2 hidden layers.

In our experiments, NNs with bump activations in their hidden layers were consistently resistant to severe overfitting while maintaining a relatively good test performance of mean AUPR at least 0.89, with a mean test AUPR always within 1 standard deviation of their counterparts with 2 hidden layers. On the other hand, NNs with RBF activations in their hidden layers maintained a high test AUPR except for halfspace separators. Meanwhile, NNs with leaky ReLU generally maintained performance within 1 standard deviation. In general, our results suggest that bump activations are more robust to overfitting while being able to achieve good performance for anomaly detection. Future work to more thoroughly investigate this overfitting phenomenon can be done to understand how to further mitigate overfitting, especially without prior exhaustive knowledge of all possible negative examples.

**Learnable $\sigma$ in bump activations**   Having a learnable $\sigma$ parameter in the hidden layers of the bump activation generally do not affect AUPR significantly for equality separator NNs optimized under LL or MSE. Moreover, a pattern emerges for both 2- and 3-hidden layered NNs – for individual NNs that

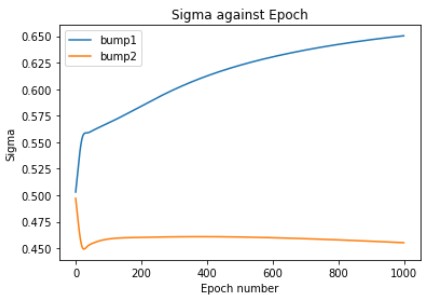
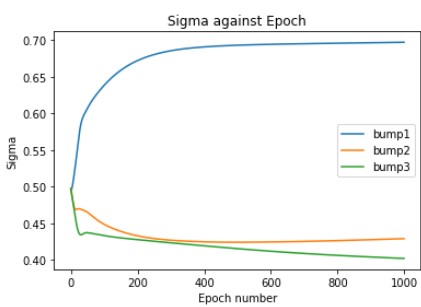

(a) Neural network with 2 hidden layers.    (b) Neural network with 3 hidden layers.

Figure 8: Example behaviour of $\sigma$ in the hidden layers of an equality separator NN with bump activations and learnable $\sigma$ in its hidden layers. The numbered suffix of the label of the curves refers to the index of the hidden layer (starting from 1).

performed well, the $\sigma$ value in the first hidden layer tends to increase while the $\sigma$ value in the latter hidden layer tends to decrease, with the final value of $\sigma$ usually being higher at the first hidden layer than the latter layers after convergence (Figure 8). A larger $\sigma$ in the earlier layer signifies allowing more information through the layer because there is a larger margin, while a lower $\sigma$ in the latter layers suggests a greater selectivity in inputs for neuron activation. Intuitively, the first hidden layer increases the dimensionality of the input for feature extraction (from 2D inputs to 5D hidden layer), while the latter hidden layers perform variable selection based on the increased dimensionality.

Such behavior of $\sigma$ is insightful to how gradient-based optimizers learn equality separation (even in hidden layers and with fixed $\sigma$). Since backpropagation through the latter hidden layers decreases the value of $\sigma$ and decreases the receptive field of the bump activation, we can infer that backpropagation is also likely to decrease the receptive field of latter layers with fixed $\sigma$ in their bump activation. This behavior corroborates with the observation in G.C. Vasconcelos (1995) that bump activations lead to more closed decision regions. Such behavior is especially useful in applications like anomaly detection, where equality separation and bump activations seems to be inclined towards encoding the inductive bias of one-class classification.

**Model averaging for increased reliability**  NNs with bump activations generally achieved high mean AUPR with some variance. Particularly, equality separators with bump activations in their hidden layers achieved the highest mean AUPR (close to perfect), for both 2- and 3-hidden layered NNs. High mean AUPR with a high variance suggest that individual NNs are able to achieve good separation of the positive and negative classes during test time. Therefore, NNs may be slightly sensitive to initializations. To mitigate potentially bad initializations, we can view these NNs as weak learners for anomaly detection. Then, an ensemble model should hopefully produce a better model for anomaly detection.

The simplest approach of averaging the predictions of the 20 models we trained achieves a perfect test AUPR for both the 2-hidden layered NN (Figure 9a) and 3-hidden layered NN (Figure 9). By averaging the prediction of the numerous trained models, the bad initializations were cancelled out and the ensemble model performs more reliably.

Other methods can be applied to ensemble the models, such as bootstrap aggregation and boosting. Future work can also look into how to initialize and optimize these NNs with bump activations well to mitigate bad initializations. Other work to directly mitigate overfitting during training of equality separators can be explored to increase the separation of normal data from anomalies to move towards a separation like the shallow equality separator with RBF kernel, such as what initializations pair well with NN optimization.

F.4 SUPERVISED ANOMALY DETECTION WITH NSL-KDD DATASET

We report additional details of experiments done in Section 4.2 for NSL-KDD dataset. Since the dataset size is significantly larger than our synthetic dataset in Section 4.1, we opt to test NNs with a

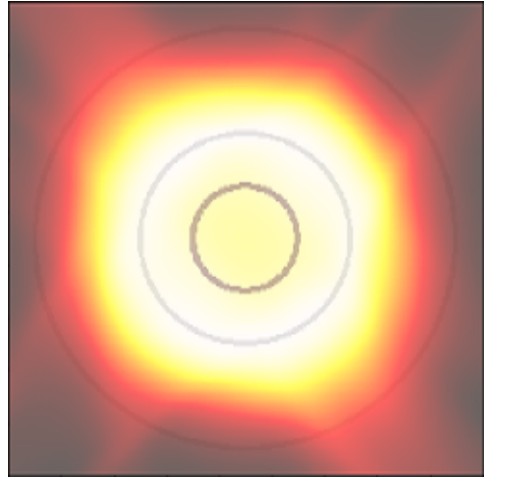 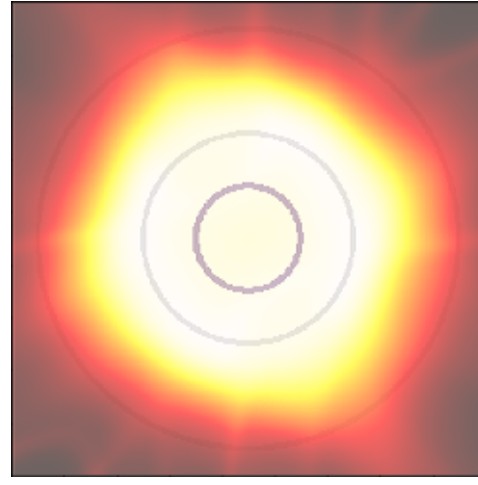

(a) 2 hidden layers.          (b) 3 hidden layers.

Figure 9: The heat map of the average of the output prediction of 20 equality separator neural networks with bump activations and fixed $\sigma = 0.5$.

homogenous type of activation function (i.e. global, semi-local or local) and repeat experiments 3 times. We report more results in Table 16.

**Unsupervised methods** For unsupervised AD methods, we test on the 2 options of dealing with labels like before: feed all the data into the algorithm, or just feed in normal data into the algorithm and remove the anomalies (because unsupervised methods assume that most/all data fed in is normal). In the main paper, we report results of only the former. Here, we report the former with a suffix "-A" and the latter with a suffix "-N" in Table 16.

We see OCSVM significantly outperforming all the other methods on the unseen anomalies for privilege escalation and remote access attacks. However, OCSVMs are unable to capitalize label information, and it is expected that the supervised SVM comfortably outperforms OCSVM as well. Once again, we notice a trade-off of poorer detection seen anomalies with the former and poorer detection of unseen anomalies for the latter. When all data is fed into an unsupervised method, it is intuitive that the model may view anomalous data as normal and misclassify them, leading to poorer detection of seen anomalies. However, when only normal data is fed into the unsupervised method, it is interesting to note that the resultant model is more conservative about the unseen anomalies. Particularly, we notice IsoF obtaining the best AUPR for DoS and probe attacks.

As an aside, we note that LOF is a distance-based algorithm, which is a likely reason for its failure in this high-dimensional setting. Both LOF and methods that require kernels (OCSVM) take a significantly longer time to train in these high-dimensional settings too.

**Binary classifiers** We details the hyperparameters for training our binary classifier NNs below.

1. We train NNs for 500 epochs with an early stopping patience of 10 epochs, with best weights restored. NNs were far from reaching the upper limit of 500 epochs.
2. We used the Adam optimizer with learning rate of $3 \times 10^{-4}$.
3. We use 1 hidden layer with 60 neurons, about half of the input dimension.
4. RBF separators and halfspace separators are trained with LL while equality separators are trained with MSE. In this appendix, we experiment with both LL and MSE and report results, noting that there is not much effect of the respective classifiers strengths and weaknesses. We note that equality separators trained with MSE also achieve similar performance to those trained with LL.
5. Some results were slightly sensitive, so we repeated experiments 5 times to obtain a more reliable result.

6. We use batch normalization before the activation function is applied (less RBF, where the only option is to apply it after).

7. Other hyperparameters are kept the same as our synthetic experiment (e.g. leaky ReLU for global activations, Glorot uniform initializer for weights).

Our choice was partially guided by the observation that it is easy to overfit the training data (i.e. training loss keeps decreasing while validation loss does not). This observation corroborates with learning theory of how the generalization error bound is looser with more expressive hypothesis classes. In particular, we experimented with 1 hidden layer with 60 neurons, 1 hidden layer with 200 neurons, 2 hidden layers with 60 neurons each and 2 hidden layers with 200 neurons each.

**NS** For NS, we maintain the same training regime but detail / modify the following hyperparameters.

1. The number of anomalies sampled was 10 times the size of the training dataset.

2. We trained our equality separators with MSE. Since anomalies can be anywhere, including close to the normal data, we use MSE to be more conservative about wrong classifications, given the potential label noise when artificially sampling anomalies.

**SAD** For SAD, we pre-train an autoencoder on our training data, then use the encoder as an intialization to train it for supervised AD as in Ruff et al. (2020). Training details are as stated:

1. We use encoder dimensions of 90-60 (i.e. a hidden layer of 90 neurons between the latent layer of 60 neurons and the input/output). We employ this for all architectures because we observe underfitting without the intermediate hidden layer. We also validate performance by observing training error converge to 0 when the autoencoder only has 1 hidden layer, where the hidden layer has the same number of neurons as the input/output.

2. Depending on the type of separator we test, the autoencoder employs either leaky ReLU (global), bump (semi-local) or RBF (local) activations homogenously.

3. Autoencoder and classification head is trained as per Ruff et al. (2020) with hyperparameters as previously stated.

4. L2 regularization for classification (Ruff et al., 2020) was tried with $10^{-1}$ and $10^{-6}$ weights and found to not to be sensitive. We used a weight of $10^{-1}$.

We note that we do not imply that RBFs are inappropriate for AD. In fact, its locality property makes it well-suited. However, RBFs are tricky to train (Goodfellow et al., 2015) and may require many neurons to work (Flake, 1993). Equality separators achieve a good balance between its ability to form closed decision regions and its capacity to learn efficiently.

## F.5 SUPERVISED ANOMALY DETECTION WITH THYROID DATASET

We report additional details of experiments done in Section 4.2 for the thyroid dataset. All details are the same as Section F.4 unless otherwise stated.

The thyroid dataset is a medical dataset on proper functioning of the thyroid (a gland in the neck), with 21 features and 3 classes, 1 of which is normal while the anomalous classes are hyperfunction and subnormal. For Thyroid dataset, we train on hyperfunction anomalies and test on both hyperfunction and subnormal anomalies.

1. By monitoring the validation loss (validation split of 0.15), we choose the model that does not overfit.

2. We choose 1 hidden layer with 200 neurons with batch normalization.

3. We initialize $\sigma = 7$ to increase the margin width so initializations are not as sensitive. (We also note that $\sigma = 5, 10$ produce similar results.)

4. We employ learning rate exponential decay with Adam optimizer under LL to speed up convergence, with an initial learning rate of 0.01 and decay of 0.96.

5. We train NNs for 500 epochs with early stopping patience of 50 epochs, with best weights restored.

Table 16: Test time AUPR for NSL-KDD. Negative sampling is denoted with a suffix (-NS). Overall AUPR against all attacks is also reported. Suffix for unsupervised AD methods determines if only normal data is used (-N) or all data is used (-A).

| Model\Attack | DoS | Probe | Privilege | Access | Overall |
|---|---|---|---|---|---|
| Random | 0.435 | 0.200 | 0.007 | 0.220 | 0.569 |
| SVM | 0.950±0.000 | 0.626±0.000 | 0.003±0.000 | 0.123±0.000 | 0.894±0.000 |
| OCSVM-N | 0.835±0.000 | 0.849±0.000 | 0.382±0.000 | 0.745±0.000 | 0.920±0.000 |
| OCSVM-A | 0.779±0.000 | 0.827±0.000 | **0.405±0.000** | **0.760±0.000** | 0.897±0.000 |
| IsoF-N | **0.964±0.006** | **0.960±0.003** | 0.039±0.007 | 0.438±0.015 | **0.957±0.002** |
| IsoF-A | 0.765±0.073 | 0.850±0.066 | 0.089±0.044 | 0.392±0.029 | 0.865±0.031 |
| LOF-N | 0.759±0.000 | 0.501±0.000 | 0.046±0.000 | 0.451±0.000 | 0.824±0.000 |
| LOF-A | 0.495±0.000 | 0.567±0.000 | 0.039±0.000 | 0.455±0.000 | 0.718±0.000 |
| HS | 0.944±0.016 | **0.739±0.018** | 0.016±0.006 | 0.180±0.033 | 0.877±0.010 |
| ES (ours) | **0.970±0.006** | 0.726±0.093 | **0.047±0.017** | **0.446±0.124** | **0.939±0.014** |
| RS | 0.356±0.002 | 0.213±0.002 | 0.010±0.000 | 0.228±0.002 | 0.406±0.001 |
| HS, MSE | 0.921±0.015 | 0.682±0.017 | 0.010±0.006 | 0.133±0.008 | 0.846±0.008 |
| ES, MSE (ours) | **0.974±0.001** | **0.717±0.107** | **0.045±0.014** | **0.510±0.113** | **0.941±0.014** |
| RS, MSE | 0.356±0.000 | 0.215±0.000 | 0.010±0.000 | 0.231±0.000 | 0.406±0.000 |
| HS-NS | **0.936±0.001** | 0.642±0.030 | 0.006±0.000 | 0.139±0.001 | 0.845±0.006 |
| ES-NS (ours) | **0.945±0.009** | **0.659±0.013** | **0.023±0.011** | 0.206±0.013 | **0.881±0.007** |
| RS-NS | 0.350±0.002 | 0.207±0.002 | 0.009±0.000 | **0.223±0.002** | 0.401±0.002 |
| HS-SAD | 0.955±0.003 | 0.766±0.011 | **0.100±0.000** | 0.447±0.000 | 0.935±0.007 |
| ES-SAD (ours) | **0.960±0.002** | **0.795±0.004** | 0.047±0.002 | **0.509±0.009** | **0.952±0.002** |
| RS-SAD | 0.935±0.000 | 0.678±0.000 | 0.022±0.000 | 0.142±0.000 | 0.846±0.000 |

Table 17: AUPR for thyroid by diagnosis (i.e. anomaly type).

| Diagnosis | Logistic Regression | HS | ES (ours) | RS |
|---|---|---|---|---|
| Hyperfunction (Seen) | 0.889±0.000 | 0.907±0.013 | **0.934±0.004** | 0.030±0.002 |
| Subnormal (Unseen) | 0.340±0.001 | 0.459±0.040 | **0.596±0.007** | 0.082±0.002 |
| Overall | 0.530±0.001 | 0.615±0.032 | **0.726±0.004** | 0.103±0.001 |

We report more detailed results in Table 17, which includes a logistic regression baseline. We see that equality separation is the best at detecting both seen and unseen anomalies.

### F.6 SUPERVISED ANOMALY DETECTION WITH MVTEC DATASET

We report additional details of experiments done in Section 4.2 for MVTec defect dataset. All details are the same as Section F.4 unless otherwise stated.

Each object has about 300 training samples, so we use (frozen) DINOv2 (Oquab et al., 2023) to embed raw image data into features rather than training our own computer vision model from scratch. DINOv2 is a foundation Vision Transformer (ViT) model pre-trained on many images, so we assume that the feature representations obtained from the model are useful, or at least more useful than a model trained from scratch. Specifically, we use the classification CLS token embedding, since it is not patch-dependent and is widely used for downstream classification tasks.

There are far fewer data samples than the ambient dimensionality of the data, so we opt for simple methods. Our halfspace, equality and RBF separators are all output-layered (i.e. no hidden layers) and act as linear probes, except the RBF separator which acts like SAD where DINOv2 plays the role of the encoder (of a pre-trained autoencoder) in Ruff et al. (2020) to extract meaningful features from the data. Hence, the halfspace separator acts as our baseline. In high dimensions, sampling to cover the whole space is much harder and computationally heavier, so we exclude negative sampling (NS).

Each anomaly class has very few anomalies (usually around 10), so we use all anomalies from 1 anomaly type during training. In other words, we do not test on seen anomalies and only test on unseen anomalies. As the best proxy (or at least an upper bound) for performance, we monitor the

training loss and AUPR as well. For instance, we observe that RBF separators do not converge and have low AUPR, which suggests that test AUPR on seen anomalies is likely to be worse. Meanwhile, the two other models (halfspace and equality separators) had converging losses and good train AUPR.

