# OpenReview forum: "Revisiting Non-separable Binary Classification and its Applications in Anomaly Detection"
_ICLR.cc/2024/Conference — Submitted to ICLR 2024_

### Official Review · Reviewer_7KS5 · 2023-10-27

**Soundness:** 2 fair
**Presentation:** 1 poor
**Contribution:** 1 poor
**Rating:** 3
**Confidence:** 3

**Summary:**

The standard linear SVM fails in the classification of the XOR problem.  To resolve this problem, the authors proposed a new paradigm, equality separation. Additionally, they integrated the idea of equality separation into the neural network and applied the proposed method to supervised anomaly detection tasks.

**Strengths:**

* The authors studied the VC dimension of the proposed equality separation. The idea of equality separation can also be applied in the neural network.
* The authors introduced the notion of closing numbers to quantify the difficulty of forming a closed decision boundary.

**Weaknesses:**

* Many times, I got overwhelmed and distracted by the narration and the layout of the manuscript. For examples,
  * The sentence "Equality separators use the distance from a learnt hyperplane as the classification metric" is confusing: A distance is measured between two objects, but here the authors only mention one object, i.e., the learnt hyperplane.
  * I feel the authors spent too much tutorial-like narration for VC dimension, closing number, and locality, in the main manuscript. In particular, what is the purpose of the over-detailed VC dimension? What useful information can we conclude in this section? Did the authors want to use the VC dimension to give some theoretical bound of generalization errors?
  * In section 2.2, it is difficult to follow the mixed descriptions. The authors may use bullet points to describe case-by-case and use some plots to support the narration, if necessary.
  * In section 3, the popped sentence "The utility of equality separators becomes evident in AD setting" is confusing since there is no particular interpretation of the utility of the equality separator in Anomaly Detection in the previous section after the introduction. As for the "Anomaly detection" in the introduction, it looks more like related works, maybe the authors could consider moving that part into Section 3.

* For the equality separator, what is the necessity of this proposed method?
  * Even though linear SVM fails to solve the XOR classification while equality separator can, why do not use kernel SVMs?
  * In Figure 3(e), what if the unseen classes fall into the purple region but are far away from the brown points? Will they be classified as brown classes when using $\epsilon$-error separator?  What if the brown class is surrounded by the blue class which consists of several cohorts? In this case, does $\epsilon$-error separator work?
  * When considering the toy example in Figure 3, the authors also use the kernel to improve the shallow equality separator. Does this imply that the proposed equality separator (even though it is simple and linear) in general is not proper without kernel or activation?

* The decision of  $\epsilon$-error separator depends on the value of $\epsilon$, but I cannot see any discussion on the choice or computation for the value of  $\epsilon$.

* Since the paper is titled "in Anomaly Detection", it should contain more well-established anomaly detection benchmarks (http://odds.cs.stonybrook.edu)

* There is no discussion on Deep One-Class Classification [1] or a comparison with it. This related work also targets anomaly detection by forming a circle boundary to the normal classes.

* Is that possible to graphically show the closed decision boundaries on other examples formed by the proposed method?

[1] Lukas Ruff, Robert Vandermeulen, Nico Goernitz, Lucas Deecke, Shoaib Ahmed Siddiqui, Alexander Binder, Emmanuel Müller, Marius Kloft Proceedings of the 35th International Conference on Machine Learning, PMLR 80:4393-4402, 2018.

**Questions:**

* Multiple minor issues:
  * Line 1, Page 3: do you mean $\mathbb{R}^+\cup\{0\}$?
  * Theorem 2.3.: separators $\mathcal{H}$ "in" Def.
  * Corollary 2.4: do you mean $\mathcal{H}_\epsilon$?

* "modeling a halfspace separator …. with an equality separator requires non-linearity like ReLU":  could the authors explain more about how the ReLU reflects the modeling for a halfspace with an equality separator?

* "equality separators yield a different optimal hyperplane compared to SVMs in binary classification": could the authors articulate the "optimality" here?

* "where equality separation is more conservative in classifying the positive brown class": What do you mean by "more conservative"?

**Details Of Ethics Concerns:**

No.

---

> ### Author Response · Authors · 2023-11-17
> **Clarifying questions and other questions**
>
> We thank this reviewer for the feedback on how to improve our paper. We updated the paper to increase its clarity based on the feedback given (e.g. on distance, bullet points, related works section, small typos). We proceed to answer other questions.
>
> # Why VC dimension?
>
> As mentioned in the global response, VC dimension analysis shows that equality separation (ES) is more expressive than halfspace separation (HS), not just a linear solution to XOR. The generalization bound can be directly obtained with VC dimension if desired.
>
> # Tutorial-like narration
>
> We explain what locality is to motivate the need for quantitative metrics (closing numbers). It was our goal to explain to a wider audience who may not be familiar with both learning theory and anomaly detection (AD) literature. If this presentation is problematic, we ask this reviewer to suggest how they think we should edit the presentation.
>
> # Why ES? Why not kernel SVMs? Can ES work without kernels?
>
> XOR is our starting point: we point out that the widely known “fact" that XOR cannot be classified with linear models is indeed false. No kernels needed.
>
> In general, the motivation of using any classifier depends on the use case. In our work, we observe interesting properties of ES for AD and corroborate this in our experiments. For instance, we observe that kernel ES perform much better than kernel SVMs in our synthetic supervised AD experiment – it is ES that performs well, the kernel merely does feature embedding.
>
> We do not over-claim that (linear) ES works for every task: raw data may not be linear classifiable. We use standard methods to embed raw data into feature space before linear classification: RBF kernels and neural networks in tabular datasets, and a pre-trained computer vision model (DINOv2) for MVTec defect dataset.  What we claim is that the _paradigm_ of ES is better for AD, that we should impose one class (the normal class) to have more structure than the other (the anomaly class). We show this theoretically with closing numbers and empirically with supervised AD experiments.
>
> # Failure modes? E.g. Fig 1e.
>
> If unseen anomalies fall in the margin, they will indeed be misclassified. Hence, the need for finite-volumed decision regions is important, and we may not be able to achieve that with just 1 ES. This observation motivates our introduction of closing numbers (especially when paired with neural networks, where neurons in a hidden layer work together), which balances the learnability and the openness of the decision region (Figure 2).
>
> Although ES may not classify everything correctly, we observe that it can classify the brown class _better_ than the corresponding HS: the set of misclassified non-brown points from ES is a proper subset of misclassified non-brown points by HS.
>
> # Discussion on $\epsilon$?
>
> The margin width is $\frac{2\epsilon}{||\mathbf w||_2}$. In Appendix F1 and F3 (Figure 8), we discuss with respect to the reparameterization of $\epsilon$ to $\sigma$ in the bump function.
>
> Our main goal was to assess the separation of normal data from anomalies – if there is no/little separation, choosing a threshold would be moot. Hence, we use AUPR as our metric. During inference, the (output) threshold depends on the tolerance on the false positive rate. Since the
> tolerance is domain-dependent, we do not prescribe a threshold in our general framework.
>
> # AD Benchmarks
>
> NSL-KDD is an improved version of the KDDCUP99 dataset mentioned in the website given by this reviewer — it is a standard cyber-security benchmark. In this revision, we included the standard MVTec defect dataset used in AD (for instance, in Explainable Deep One-Class Classification [1]) and thyroid dataset from the suggested website.
>
> # Deep One-Class Classification (Deep SVDD) discussion?
>
> In our paper, we use the improved Deep Semi-supervised Anomaly Detection (SAD) as a SOTA benchmark, which is Deep SVDD that can use labels. This discussion is in section 3.1 and 4.2 e.g. “SAD is … an RBF separator…”.
>
> # Possible to graphically show other examples?
>
> We are unsure of what this reviewer means and would like to ask for clarification.
>
> # Modeling halfspace with ES and ReLU?
>
> As per section 2.2, ES acts as a HS by splitting the space into the positive and the zero parts of the ReLU.
>
> # Optimality of 2 ES for HS?
>
> We answer this in the last 2 sentences of section 2.2 from a robust perspective. Support vectors may be outliers far from high density regions, so using them may not be as robust as fitting a hyperplane for each class (which depends more on the center of mass).
>
> # Meaning of “conservative”?
>
> A more conservative classifier is less likely to classify a datum as positive, in line with density level set estimation. Closing numbers is a property of a class of classifiers that quantify conservativeness, motivated by using these classifiers in neural networks.
>
> [1] Liznersk et al., Explainable Deep One-Class Classification. ICLR 2021.

---

### Official Review · Reviewer_NTiL · 2023-11-01

**Soundness:** 3 good
**Presentation:** 3 good
**Contribution:** 2 fair
**Rating:** 5
**Confidence:** 2

**Summary:**

The authors explore the space of halfspace separator. In this manuscript they explore the equality separator. Instead of dividing the space into two halves, all instances that fall on the hyperplane (or are near to it) are part of one class while the rest belong to the other class. The others then calculate the VC dimension of this equality separator. Furthermore, they also introduce the bump activation function to be used in NNs which is a smoothed version of the equality separator. They propose using this separator for anomaly detection. Finally, they show the efficacy of the proposed method in the experimental section.

**Strengths:**

1. The proposed equality separator is very interesting. Even though for epsilon-separator is related to SVMs there is still other novel aspects to this. Furthermore the theoretical analysis shown here for VC dimension shows the advantage of the proposed method over regular linear separators.
2. The results for anomaly detection is promising specially on the synthetic data set.
3. The paper is very well written. All required information is provided in a clear manner and explained properly.

**Weaknesses:**

1. As mentioned above, the anomaly detection results in this paper are promising. However, the gain on the NSL-KDD dataset is not always positive. This limits the application of the proposed method.
2. The authors performed thorough experiments on the NSL-KDD dataset. However, further datasets should also be included in the experimentation to show the efficacy of the proposed method.

**Questions:**

1. This is related to concern regarding weakness 1. What are the authors intuition regarding the equality separator not always outperforming the other baseline methods for NSL-KDD.
2. I noticed that in Table 3, for DOS, HS-NS result is in bold. Why is that? I though ES-NS performs the best here?

---

> ### Author Response · Authors · 2023-11-17
> **Question on Performance**
>
> We thank this reviewer for affirming the creativity of equality separation, the theoretical analysis provided and the communication of our ideas. We acknowledge this reviewer’s request for more datasets in our global response (adding MVTec and thyroid dataset results), and thank this reviewer for noting the thorough experimentation on NSL-KDD dataset. We address other concerns below.
>
> # Why don’t equality separators have the best performance for NSL-KDD dataset in all metrics, especially over (unsupervised) baseline methods? Are there intuitions why?
>
> In short, equality separation is an attempt to merge the benefits of unsupervised methods into supervised methods. We observe a _significant_ improvement over supervised methods for unseen anomalies while maintaining supervised-level performance on seen anomalies.
>
> We do not claim that equality separation is or should be the _final_ solution to supervised anomaly detection (AD). We do claim that, among methods that use label information via standard empirical risk minimization, equality separation is more suitable, especially to detect unknown anomalies. The experiments support this claim: for instance, in regular binary classification, equality separation _significantly_ improves AUPR on detecting unseen anomalies (privilege and remote access attacks) by 3-fold compared to halfspace separation.
>
> Comparing equality separation to unsupervised AD baselines, we especially see that OCSVM beats all other methods in detecting unseen anomalies (privilege and remote access attacks). **However**, unsupervised methods like OCSVM cannot capitalize on labels, hence our comment in footnote 5 that unsupervised methods are not our main apples-to-apples comparison. The inability to use labels impacts detection on seen anomalies (DoS attacks). This disadvantage is in addition to the computational load of computing kernels (e.g. for OCSVM) for large datasets like NSL-KDD. Furthermore, equality separation outperforms unsupervised baselines Isolation Forest and Local Outlier Factor on the unseen remote access attacks, while other regular binary classifiers are far from it. We conclude that equality separation is a step in the right direction to use limited anomaly labels to detect **both seen and unseen** anomalies.
>
> Our intuition on why equality separation may not be beating the OCSVM baseline in detecting unseen anomalies relates mainly to implementation details – given that equality separation had excellent performance in our synthetic experiments, we posit that higher dimensional data amplify certain difficulties. We observe that optimization may be more difficult because of more sensitivity in initializations and vanishing gradients. We are currently working on understanding how the activation function, loss function and optimization scheme (including initializations) work together to form closed decision regions, which we posit may be the key to further improvements.
>
> # Bold Results for HS-NS in Table 3?
>
> We consider both the mean and standard deviation when determining which is the best. This is to account for variability in running the optimization that arises from different initializations. To avoid confusion, we account for this consideration in our discussion and appendix instead and remove the double bold text in the table in the main body.

---

> > ### Comment · Reviewer_NTiL · 2023-11-22
> > **Response to author rebuttals**
> >
> > I appreciate the authors for providing the rebuttal. However, after reading the rebuttal and the other reviews, I still think that there are some weaknesses in this manuscript. Thus I have decided not to update my score at this time.

---

> > > ### Author Response · Authors · 2023-11-22
> > > **Response to reviewer**
> > >
> > > We thank this reviewer for dedicating their time to review our work and for their gracious comments on the paper's interesting approach coupled with theoretical VC dimension analysis, promising experimental results and well-written communication.
> > >
> > > In our responses (global and reviewer-specific), we have addressed the concerns and questions this reviewer raised in their initial review. Given the positive aspects this reviewer highlighted in their initial review, we are eager to understand specific areas that this reviewer believes may need further attention or improvement. These insights are valuable to us in refining the clarity, motivation and overall quality of the paper.
> > >
> > > We will carefully consider all the feedback provided by reviewers to ensure a comprehensive revision that addresses various perspectives. Once again, we appreciate this reviewer's time and continued engagement in reviewing our work.

---

### Official Review · Reviewer_FBaF · 2023-11-01

**Soundness:** 4 excellent
**Presentation:** 3 good
**Contribution:** 3 good
**Rating:** 6
**Confidence:** 2

**Summary:**

This work discusses a novel approach to linearly classify the XOR problem, challenging the conventional wisdom that it cannot be done with linear methods. The authors propose "equality separation" as an alternative to traditional halfspace separation, adapting the SVM objective to distinguish data within or outside the margin. They integrate this classifier into neural network pipelines, highlighting its potential for anomaly detection and demonstrating its effectiveness in supervised anomaly detection experiments, including detecting both seen and unseen anomalies.

**Strengths:**

- The introduction of an 'equality separator' to address the XOR problem is indeed an intriguing and innovative concept.
- The introductory section is well-structured and easily comprehensible, complemented by the informative Figure 1.
- All the theoretical assertions are substantiated with precise definitions and rigorous proofs.

**Weaknesses:**

- In order to enhance the accessibility and comprehensibility of the content, it would be advisable to incorporate critical discussions and analyses that are currently relegated to the appendix into the main body of the manuscript.
- The proposed design has exclusively undergone experimentation on toy datasets or relatively straightforward real-world datasets. Consequently, there is uncertainty surrounding the effectiveness of the proposed method when confronted with more intricate, real-world datasets.

**Questions:**

- Is this network design extensible to more intricate datasets, such as image data?
- Isn't there a gradient vanishing problem with that bump activation design when the layers of the neural network are deep?
- What are the advantages of doubling the VC dimension in contemporary neural network architecture?

---

> ### Author Response · Authors · 2023-11-17
> **Questions on Dataset, Gradients and VC Dimension**
>
> We thank this reviewer for appreciating the innovation of equality separation, effective visualizations and theoretically sound analysis. We address this reviewer’s question on general applicability (to other problem domains like computer vision) in our global response (adding MVTec and thyroid dataset results). We address other questions here.
>
> # There are some useful sections in the appendix that would benefit to be in the main body.
>
> We thank this reviewer for reading through the details in the appendix. We are willing to move relevant sections of the appendix to the main paper. Which sections does this reviewer think are relevant to move to the main body?
>
> # The proposed method is tested only on straightforward datasets. Can it be generalized to other datasets such as images?
>
> Although we use the toy dataset for enhanced visualizations and understanding (both in the main body and appendix), we use NSL-KDD as a real cyber-attack dataset. NSL-KDD is an (improved) edited version of the KDDCUP99 dataset, which was part of USA DARPA’s Intrusion Detection System evaluation program, with “about 4 gigabytes of compressed raw (binary) tcpdump data of 7 weeks of network traffic, which can be processed into about 5 million connection records, each with about 100 bytes. The two weeks of test data have around 2 million connection records” [1]. Real cyber-attack datasets allow us to test methods against adversaries who are actively trying to circumvent and break the system.
>
> Our method can indeed be generalized to other domains – there is nothing specific to tabular or cyber-attack data. We have included results for MVTec image defect dataset, showing better AUPR in 5/6 objects tested. We also experiment with the thyroid dataset, which focuses on a medical application. Our method once again shows better performance in detecting seen and unseen anomalies.
>
> # Will gradients vanish in deep networks?
>
> Gradients will vanish outside the margin (proportional to $\frac{1}{||\mathbf w||_2}$) as plotted in Figure 4 (Appendix E1), similar to vanishing gradients in sigmoid or the dead neuron problem in ReLU. However, like sigmoid, gradients are the greatest in the regions close to the boundary between positive and negative (i.e. regions of aleatoric uncertainty).
>
> There are 2 sides of this coin. The conventional perspective is that gradients that vanish inhibit learning, and we mention in Section 4 and Appendix F1 that we can mitigate this by initializing $\sigma$ to be large (e.g. $\sigma=10$). This decreases the sensitivity in weight initialization.
>
> On the flip side, there is a self-regularizing effect, where the neuronal activation is very selective. This is indeed what we desire in anomaly detection, where unseen inputs should not activate the network. In Appendix F3 (Figure 8), we show in Figure 8 how the margin width changes across hidden layers (grows across time in earlier layers for more information flow and shrinks in later layers for selectivity of information/features).
>
> # Advantages of doubled VC dimension in contemporary neural network architecture?
>
> As mentioned in our global response, the VC dimension analysis was not aimed at a neural network analysis. It shows that equality separation is indeed more expressive than halfspace separation (i.e. not that equality separation works for XOR and nothing else).
>
> In neural networks trained for anomaly detection, we champion for using the equality separation paradigm, which has an inductive bias towards closed decision regions while maintaining the capacity to learn. We think of closing numbers to anomaly detection and forming closed decision regions as VC dimension is to risk minimization – closing numbers is a dimension-like combinatorial number describing the capacity of a hypothesis class to minimize open space risk. Closing numbers also have a direct interpretation in a neural network setting, as we described in section 3.3.
>
>
> [1] M. Tavallaee, E. Bagheri, W. Lu and A. A. Ghorbani, "A detailed analysis of the KDD CUP 99 data set," 2009 IEEE Symposium on Computational Intelligence for Security and Defense Applications, Ottawa, ON, Canada, 2009, pp. 1-6, doi: 10.1109/CISDA.2009.5356528.

---

### Author Response · Authors · 2023-11-17
**Global Response: Higher VC dimension means more expressive, 2 more benchmark datasets included**

We thank all reviewers for their observations on the innovation of equality separation, theoretical analysis (e.g. VC dimension) and for other helpful comments. We address common questions here and summarize this response as follows:

1. Higher VC dimension of equality separation shows that it is not just limited to solving XOR, but is **generally more expressive**.

2. We note that NSL-KDD dataset is a real DARPA cyber-attack evaluation dataset, but we also **provide additional results on other datasets** (Table 4 and below for reviewers' ease) as per requested.

# Higher VC Dimension

First, we show that the VC dimension of equality separators is larger than halfspace separators for the following reason: it is not merely XOR that equality separators are better suited for than halfspace separators, but that equality separators are indeed more expressive than halfspace separators _in general_. Linear classifiers are not bound to have a VC dimension of $n+1$.

We explore the implications of these in anomaly detection in Section 3.4. By asserting that normal data should be within the margin (i.e. labeled as the positive class), we disallow label flipping, so (modified) equality separators have the same VC dimension as halfspace separators. Here, 2 errors are controlled:

1. Through reduction in VC dimension, we improve estimation error.

2. With domain knowledge of normal versus anomalies, we also reduce approximation error (through what is commonly known as _open space risk_ in OOD detection/open set recognition [1]) by specifying that normal data is more constrained (or, more precisely, that equality separators reduce open space risk).

We clarify this in the revised paper.

# Added 2 benchmark datasets

Second, we thank one reviewer for noting the thorough experimentation on the NSL-KDD dataset and effective visualizations on the 2D dataset. We note that reviewers would also like to see if equality separation is more generally applicable. Hence, we included the classic MVTec defect dataset to experiment with images in a manufacturing application, and thyroid dataset for a medical application. We note that the cyber-attack NSL-KDD dataset presents scenarios of anomalies from malicious human adversaries, MVTec presents anomalies that are complicated due to the high dimensions, while thyroid is a medical anomaly . For MVTec, equality separation is best in 5/6 of the objects we test. For thyroid, equality separation detects seen and unseen anomalies the best. We add these results in Table 4 of the revised paper.

_Note_

As a note, the paper has also been edited for clarity (based on reviewers’ comments) and for conciseness to ensure that the paper stays within 9 pages. The main edits are in Section 2.2 (clarity) and Section 4.2 (include thyroid and MVTec defect dataset experiments and results in Table 4), with additions of Appendix F5 and F6 for details on thyroid and MVTec experiments respectively.

We once again thank reviewers for their time and the helpful comments on the communication of the paper. We hope that we have sufficiently addressed all concerns through edits or responses.

[1] W. J. Scheirer, A. de Rezende Rocha, A. Sapkota and T. E. Boult, "Toward Open Set Recognition," in IEEE Transactions on Pattern Analysis and Machine Intelligence, vol. 35, no. 7, pp. 1757-1772, July 2013, doi: 10.1109/TPAMI.2012.256.

# Results

*Table 4a/17. AUPR for thyroid dataset by diagnosis (i.e. anomaly type).*
|Diagnosis|Logistic Regression|HS|ES (ours)|RS|
|:-|:-:|:-:|:-:|:-:|
|Hyperfunction (Seen)|0.889±0.000|0.907±0.013|**0.934±0.004**|0.030±0.002|
|Subnormal (Unseen)|0.340±0.001|0.459±0.040|**0.596±0.007**|0.082±0.002|
|Overall|0.530±0.001|0.615±0.032|**0.726±0.004**|0.103±0.001|

*Table 4b. Overall AUPR for MVTec dataset in detecting unseen defects (i.e. anomalies).*
|Object| HS | RS (SAD) | ES (ours) |
|:-|-:|-:|-:|
|Capsule|0.918±0.002|0.894±0.000|**0.947±0.003**|
|Hazelnut|**0.932±0.006**|0.783±0.000|**0.932±0.034**|
|Leather|**1.000±0.000**|0.848±0.000|0.996±0.001|
|Pill|0.917±0.005|0.908±0.000|**0.933±0.014**|
|Wood|0.984±0.004|0.866±0.000|**0.986±0.007**|
|Zipper|0.998±0.000|0.879±0.000|**1.000±0.000**|

*Table 4c. AUPR for pill object in MVTec dataset, by defect (i.e. anomaly type).*
|Defect|HS|ES (ours)|
|:-|:-:|:-:|
|Combined|0.913±0.010|**0.943±0.016**|
|Contamination|0.768±0.029|**0.802±0.114**|
|Crack|0.652±0.009|**0.730±0.046**|
|Imprint|0.470±0.035|**0.573±0.094**|
|Type|0.885±0.071|**0.910±0.156**|
|Scratch|0.655±0.021|**0.690±0.058**|

\* Legend: HS is halfspace separator, ES is equality separator (ours), RS is RBF separator and SAD is Deep Semi-supervised Anomaly Detection.

---

### Meta-Review · Area_Chair_cUon · 2023-12-08

**Metareview:**

The authors challenge the standard binary classification model by switching to equality-based decisions that are usually present in regression scenarios. Thus, the motivation of the paper can either be seen as cleverly bridging the gap between the two worlds or violating traditional assumptions (e.g., classification is usually considered as tipping the instances over the decision boundary rather than exactly predicting their label, logistic regression perhaps being an exception). Hence, the paper begins with a somewhat bold claim and uses the idea of 'equality separation' to devise a novel approach for anomaly detection. However, the implications of the 'equality separation' are unclear and should be investigated more carefully. Connections to (logistic) regression and other standard settings are theoretically not explored.

**Justification For Why Not Higher Score:**

In my view, the writing was a bit too bold for the reviewers and too much space was dedicated on (outdated) empirical results rather than theoretical insights. The implications of the 'equality separation' (the wording is perhaps sub-optimally chosen) are unclear and should be investigated more carefully. Connections to regression settings are not explored in sufficient detail.

**Justification For Why Not Lower Score:**

N/A

---

### Decision · Program_Chairs · 2024-01-16

Reject